# Metacognition biases information seeking in assessing ambiguous news
Valentin Guigon[1,2], Marie Claire Villeval [2,3,4] & Jean-Claude Dreher [1,4] ✉

How do we assess the veracity of ambiguous news, and does metacognition guide our decisions to seek further information? In a controlled experiment, participants evaluated the veracity of ambiguous news and decided whether to seek extra information. Confidence in their veracity judgments did not predict accuracy, showing limited metacognitive ability when facing ambiguous news. Despite this, confidence in one's judgment was the primary driver of the demand for additional information about the news. Lower confidence predicted a stronger desire for extra information, regardless of the veracity judgment. Two key news characteristics led individuals to confidently misinterpret both true and fake news. News imprecision and news tendency to polarize opinions increased the likelihood of misjudgment, highlighting individuals' vulnerability to ambiguity. Structural equation modeling revealed that the demand for disambiguating information, driven by uncalibrated metacognition, became increasingly ineffective as individuals are drawn in by the ambiguity of the news. Our results underscore the importance of metacognitive abilities in mediating the relationship between assessing ambiguous information and the decision to seek or avoid more information.

The unprecedented growth of the internet and social media platforms has been accompanied both by an abundance of content and by the spread of misinformation[1–4]. Misinformation, characterized by false, inaccurate, or misleading information, yields devastating consequences at the societal level, fueling polarization and fostering resistance to crucial initiatives such as climate action and vaccination efforts[5–7]. Contrary to disinformation, misinformation does not need to be created deliberately to mislead. The inherent imprecision of misinformation frequently blurs the perceived boundaries between true or false information. Moreover, its capacity to create an illusion of consensus can inadvertently undermine individuals' ability to discern information authenticity. The appearance of credibility stemming from unified perspectives often obstructs critical evaluation, leaving individuals susceptible to unwittingly compromising their assessment of information authenticity[8]. On the contrary, content that appears to divide opinions may create an illusion of unlikeliness, encouraging people to dismiss the information. These two characteristics, imprecision, and polarization, increase ambiguity[9,10], amplifying the difficulty to discern between true and false information. Individuals not only face challenges when having to evaluate the veracity of the information they are exposed to, but they also struggle to effectively search for extra information to verify social and political claims, sources, and evidence[11]. A number of strategies have been proposed to prevent the spread of misinformation[12–14], including fact-checking, directing attention to accuracy[15–17], censorship, encouraging

more selective sharing by individuals[18,19], or capping the number of others to whom messages can be forwarded[20]. Yet, fact-checking at the speed and scale of today's platforms is often impractical for private companies or government agencies[20]. An alternative approach could involve targeting individuals themselves and focusing on enhancing their abilities to assess the veracity estimation[21,22].

Here, we were interested in understanding the cognitive mechanisms and the relationships between individuals' judgment about the veracity of ambiguous news (news for which the probability of being true is unknown) they are exposed to, the confidence in such judgment, and the willingness to seek additional information to better assess news veracity.

The willingness to gather extra information rather than sticking to one's current knowledge may depend critically on the subjective confidence in one's assessment. This has been demonstrated in the domain of perceptual decision-making[23,24] but remains to be studied regarding real news. In perceptual decision tasks, confidence in one's judgment accuracy plays a direct and determinant role in one's willingness to sample more evidence to update one's beliefs[24–27]. In those tasks, subjective confidence in one's judgment accuracy correlates closely with objective accuracy[28–32]. However, in real-world situations, such as assessing the veracity of media news, the role of confidence in information search remains unclear, as is the relationship between subjective confidence and objective accuracy in the assessment of news veracity. Because decision accuracy and confidence are

[1]Neuroeconomics lab, Institut des Sciences Cognitives Marc Jeannerod (ISCMJ), CNRS UMR 5229 and Université Claude Bernard Lyon 1, Bron, France. [2]CNRS, Université Lumière Lyon 2, Université Jean-Monnet Saint-Etienne, emlyon business school, GATE, Lyon, France. [3]IZA, Bonn, Germany. [4]These authors contributed equally: Marie Claire Villeval, Jean-Claude Dreher. ✉e-mail: dreher@isc.cnrs.fr

typically highly correlated[24,33,34], it is difficult to identify whether confidence causally influences the demand for extra information. Crucially, to provide such evidence concerning real news, one needs to demonstrate that confidence in one's judgment predicts the demand for extra information while controlling for objective performance accuracy in judging news as true or false. To that purpose, we designed our experiment using ambiguous news contents, deliberately leading to performance accuracy at chance level when assessing the veracity of this news. Understanding these relationships requires experimentalists to meticulously select news and rigorously control for their level of ambiguity.

Our focus was on news reflecting ambiguous information that could be either false or true, with varying levels of perceived ambiguity. Importantly, such ambiguous news was sourced from agents with no intention to deceive, thereby ruling out a deliberate willingness to propagate fake news. Examples of naïve agents endorsing and spreading false inaccurate information with no intention to deceive abound[35]. Recent research reports that only a very small percentage of people purposely endorse sharing misinformation online[36]. We designed an incentivized within-subject experiment in which non-ego relevant news varied in content imprecision and propensity to polarize opinions. Specifically, headlines only concerned non-partisan news, that is, news unrelated to political parties. We introduced variations of ambiguity in stimuli at the time of designing the task. Ambiguity was subjectively assessed by a separate sample of 55 participants during a pre-testing. Participants were presented with a set of brief news about ecology, democracy, and social justice taken from the press that could be either true or false. Participants had to evaluate the veracity of each brief news and report their confidence in their judgment on a continuous scale, using a probability elicitation incentivized method. Then, participants had to decide on whether acquiring or not additional information about this news (to be received after the task was performed), and report their willingness-to-pay to have their information-seeking choice implemented (Fig. 1). Importantly, this latter procedure ensured that the willingness to acquire or not acquire extra information is equally balanced. This is unlike previous procedures used in the perceptual decision-making domain for which acquiring extra information was costly but for which not acquiring extra information was free (e.g.,[25]).

We investigated the relationship between objective accuracy in judging news veracity and confidence in this judgment, controlling for the role of news ambiguity in the judgments of news as true or false. Specifically, we first tested whether confidence in one's judgment about news veracity predicts success in judging news veracity. Under varying ambiguity, we anticipated uncalibrated metacognition, with participants' confidence uncorrelated with actual success. We then tested whether such confidence in one's judgments drives the demand for extra information about the news. In line with the perceptual decision-making literature on confidence-based information-seeking[24,25,37], we predicted that the demand for extra information should increase when confidence in one's veracity assessment is at its lowest. In our context, confidence rather than beliefs is anticipated to play a pivotal role. Finally, we performed a moderated mediation analysis to reveal the relationships between the mechanisms underlying judgments of news veracity and the mechanisms underlying the demand for extra information about the news.

## Methods

### Participants

269 participants with no history of neurological or psychiatric disorders participated in this online experiment run on Testable.org. Data were collected in two waves. The first one took place with 80 participants in November 2020. A second one with 189 participants spanned from December 2021 to January 2022. Except for additional questions in the final questionnaire, there were no differences in the experimental design between the two waves. Participants in the experiment were students from business schools and engineering schools regularly registered in the GATE-Lab pool of experimental subjects, at the University of Lyon, France. They were paid on average $15.92, including a $9 show-up fee, for an experiment that lasted 46 min on average. All conditions, data collection procedures, and data exclusion criteria have been transparently reported in this manuscript. Two participants were excluded from the analyses due to outlying response times ("RT") during news evaluation (one subject: mean RT = 51.79, SD = 26.35; one subject: mean RT = 1.93, SD = 1.31) compared to the mean response time (14.41, SD = 8.44). Nine participants were excluded because they did not complete the final questionnaire. All collected data and questionnaires have been reported in either the manuscript or the Supplementary Information. A total of 258 participants, aged 18–34 years, were included in the statistical analyses (sex: 131 female participants, 127 male participants, mean age = 21.9, SD = 2.78). Sex was collected by Testable.org as self-report measures.

The study was not preregistered. This research complies with the Declaration of Helsinki (2013), aside from the requirement to preregister human subjects research, and received approval from an internal ethics review board. It also complies with the European Data Protection Regulation (GDPR), and informed consent was obtained from all subjects prior to participation. No artificial intelligence-assisted technologies were used in this research or the creation of this article.

### Task and design

To select our stimuli, we set up a pre-test of every stimulus with independent raters and kept the stimuli that best fit our criteria (mean agreement = 4.66, SD = 1.44) (see Supplementary Methods I). The resulting dataset had an average success rate, as calculated per stimulus, of 51.58% (SD = 20.24%).

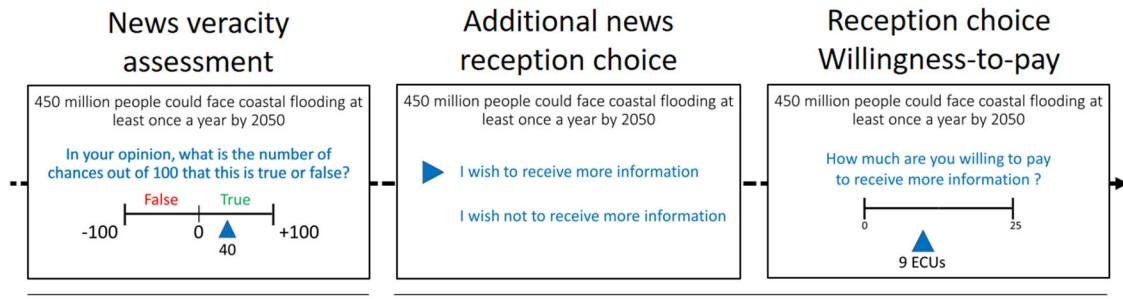

**Fig. 1 | Description of the task.** Participants read brief news and were incentivized to report the probability that the news was true or false, allowing us to assess both veracity judgment and confidence in one's judgment. A correct evaluation of news veracity (i.e., true news judged as true and false news judged as false) was worth 50 ECU while an incorrect evaluation was worth 0 ECU (eight trials out of forty-eight were selected at random to be paid). Next, participants had to choose between receiving or avoiding receiving more information about the news. Given their choice, they had to indicate how much they were willing to pay (from 0 to 25 ECU) to have this choice implemented (endowment = 200 ECU, eight trials chosen at random to be implemented). A Becker–DeGroot–Marschak (BDM) procedure determined whether their choice would be, or not, implemented, and at which price, depending on their bid. This procedure ensured that both the demand to receive and the demand to avoid receiving extra information were costly.

The most difficult stimulus to evaluate had a 6.92% success rate and the easiest stimulus to evaluate had a 93.75% success rate. Stimuli included, for instance, 'Some well-known athletic committees still deliberately maintain certain sports-related discriminations', 'In 2018, greenhouse gas emissions decreased by 4% in France for the first time', and 'The most common voting system in Europe is the "preferential vote," which involves ranking candidates in order of preference'. We computed the scores of news content imprecision (M = 5.28, SD = 1.32) and propensity to polarize (M = 6.41, SD = 1.62) using Intra-Class Correlations, with 11 raters and 42 heterogeneous samples. The recommended number of raters ranges between 3, with at least 30 heterogeneous samples[38], and 20[39]. Overall, our procedure closely follows the practical guide of Pennycook and colleagues for behavioral research on fake news and misinformation[40]. In addition, we ran a sentiment analysis on all stimuli, separating for true and false news. Out of the 96 stimuli, 93.75% of the news was predominantly categorized as emotionally neutral. The distribution of false information, rated as negative by the sentiment analysis, was broader compared to the distribution of true information. Conversely, we found the opposite trend for information rated as positive (see Supplementary Methods V.1, Supplementary Fig. 1).

Individuals' worldviews have been shown to explain what they believe to be true[41]. As a proxy for beliefs, we adapted measures of social distance between individuals to the relationships individuals may maintain with organizations. In our context, a closer social distance would mean more involvement in the concerns related to the themes. To do that, as in ref. [42], we measured closeness and liking. Unlike the latter, as it may be difficult to interpret in the context of a human being's relationship to an organization, we replaced similarity with familiarity with the organization. This approach was based on the premise that the more an individual is involved in the concerns related to a particular theme, the more that individual will be aware of the different actors operating within that theme. To have a proxy of such prior beliefs we instructed participants in the first part of the experiment (Supplementary Methods II.1) to rate various political organizations that were related to the different news domains. We selected 12 organizations active in the domains of ecology, democracy, or social justice (Supplementary Methods III). Each organization was described by a 1000-character (±20%) statement taken from the organization's websites, with minimal manipulation of the original website content. Participants indicated with six responses their liking, familiarity, and closeness of values concerning organizations in direct connection with the topics of the news., on a scale from 0 to 7. For each topic, we selected two organizations aligned with concerns related to the news, and two organizations misaligned with them. Organizations were presented to participants in a randomized order.

We computed the participants' adhesion to each organization (as a proxy of the knowledge of the domain) by aggregating their six responses in a score that was normalized on a scale from 0 to 100. The higher the score, the more likely the participant was to adhere to the organization and be knowledgeable about its domain of activity. After rating the organizations, participants read the instructions on the task.

The second part of the experiment involved a two-stage task (Supplementary Methods II.2). The first stage included the veracity judgment task. Participants were divided into two groups that received 48 different stimuli each. Each of the 48 trials started with a fixation cross on the screen (Fig. 1). Then, a brief news, either true or false, was displayed. Participants were asked to report what was, in their opinion, the number of chances out of 100 that this news was true or false. Their response revealed their degree of confidence in their judgment. To respond, participants moved a slider either to the left (False) or to the right (True). The slider started at −100 on the left side and ended at +100 on the right side. Participants could not respond with 0. Thus, each move in a direction incremented their degree of confidence by 1%. The elicitation of probabilities was incentivized, following the Karni procedure[43]. This elicitation method is incentive-compatible, considered relatively easy to understand[44], and frequently used in economics[45,46]. With this elicitation method, truthful reporting is the unique dominant strategy. We adopted an approach used in previous experiments[47,48] that employs a narrative involving robots to explain the process to participants.

Participants were informed that, after the experiment, we would randomly draw eight trials and reward correct veracity judgments in these trials. To be more specific, for each selected trial, considering the participants have reported their confidence $\mu \in [1, 100]$ regarding their judgment of the news veracity, the elicitation mechanism selected a random number $r$ from a uniform distribution on [1, 100]. If $\mu \geq r$, the participants earned a payoff $\beta := 50ECU$ if their veracity judgment was correct and a payoff $\beta := 0ECU$ if their veracity judgment was incorrect. If $\mu < r$, the payoff was determined by a lottery $(r, 50ECU, 0ECU)$. Participants understood that if $r$ exceeded their reported confidence $\mu$, the outcome would rely on the lottery; otherwise, their own judgment would be used. Participants were informed that truthful reporting was in their best interest. Each correct veracity judgment in this context earned a reward of 50 ECU, with 100 ECU valued at $2. The narrative $r$ corresponded to the accuracy level of a robot randomly drawn from a pool of 100 robots.

The second stage corresponded to the elicitation of the demand to receive extra information. After validating their veracity judgment and while their screen was still displaying the brief news, participants were asked to choose between receiving or not additional information related to the same news after the completion of the experiment. Finally, they had to report how much they were willing to pay, between 0 and 25 ECU of their 200 ECU initial endowment, to have their decision implemented (i.e., to receive or not receive further information), using the Becker–DeGroot–Marschak (BDM) procedure[49]. We kept the cost constant between reception choices to compare them while controlling for scaling or anchoring effects. This aligns with literature showing that people may value ignorance and are even willing to pay not to receive information. In the case participants opted for more information, regardless of whether the information was true or false, they were eligible for receiving a debunking article investigating the content of the brief news in detail. Debunk articles were taken from the French fake news debunk platforms *Les Décodeurs du Monde*, *AFP Factcheck,* and *Libération Checknews* from the period 2017–2020. The additional information was sent by email to the participants after the experiment. Participants were informed about the mechanism and it was always possible to avoid receiving information for sure by paying the maximum amount. If participants did not pay enough to avoid receiving more information, participants received more information about the news, contrary to their decision. All these aspects were made common knowledge before participants made their choices. This choice of design was motivated by the findings in the literature that people may wish not to receive information that decreases the ambiguity they would rather maintain (i.e., valuation of ignorance[50–55]).

At the end of the experiment, we randomly selected eight trials among the 48. For each selected trial, if the participant's willingness-to-pay (WTP) was equal to or above a randomly selected price between 0 and 25 (each price had an equal probability of being drawn), the program deducted the randomly selected price from his or her 200 ECU endowment and his or her decision was implemented. If the WTP was lower than the price, no deduction was operated and the option the participant did not choose was implemented. Using a bidding mechanism such as the willingness-to-pay instead of relying on stated preferences, such as ordinal ranking (i.e., simply choosing to receive or not to receive), reduces the likelihood that participants did not provide sincere responses.

After reading the instructions on the task, the participants filled in a comprehension questionnaire about these instructions (Supplementary Methods II.3).

At the end of the experiment, participants had to fill in several questionnaires allowing us to measure notably their exposition of information and their degree of curiosity (see Supplementary Methods IV). Epistemic curiosity may respond to the desire to stimulate positive feelings of intellectual interest or the desire to reduce undesirable states of information deprivation[56]. To check the relationship between veracity assessment, the demand for further information, and epistemic curiosity, we administered the Litman questionnaire of Epistemic Curiosity[56]. Participants in the second wave of data collection answered additional questions about their perceived share of fake news circulating on the Internet and social media.

**Table 1 | The models of news content imprecision and news content polarization explain best participants' success in assessing the news veracity**

| Model comparison with WAIC | | | | | | |
|---|---|---|---|---|---|---|
| **Models** | **ΔWAIC** | **ΔSE** | **WAIC** | **SE WAIC** | **pWAIC** | **Weight** |
| News content imprecision & polarization | 0 | 0 | 16890.63 | 33.54 | 10.3 | 1 |
| News content polarization | −30.22 | 8.06 | 16951.08 | 29.88 | 8.27 | 0 |
| News content imprecision | −105.5 | 14.44 | 17101.64 | 17.33 | 8.59 | 0 |
| Response times | −125.99 | 16.96 | 17142.62 | 11.55 | 8.86 | 0 |
| Non-informative beta response | −132.22 | 16.72 | 17155.06 | 9.99 | 7.36 | 0 |
| Beliefs alignment | −134.52 | 16.37 | 17159.68 | 7 | 6.25 | 0 |
| Subject random-effects | −134.56 | 16.37 | 17159.75 | 6.69 | 6.31 | 0 |
| Veracity judgment | −135.01 | 16.34 | 17160.65 | 7.26 | 7.3 | 0 |

The table displays model comparisons ordered by WAIC. The best model has the lowest WAIC, showing the best out-of-sample capacity, and higher weight, showing the best prediction of in-sample data. Models are described in the "Data analysis" section of the "Methods" section.

The objective was to check for a potential relationship between distrust in channels of information and veracity estimations. Data on participants' exposure to information was collected but not analyzed, as it was outside this study's scope. Nationality data was collected by Testable.org as self-report measures along with sex (biological attribute assigned at birth). Gender data (shaped by social and cultural circumstances) and ethnicity data were not collected.

All the procedures, collected data, and questionnaires relative to the pre-test, including English translations of the task instructions and questionnaires, are available in the Supplementary Information (procedure: Supplementary Methods I.1, instructions: Supplementary Methods I.2). The instructions for the task and the questionnaires, translated into English, are available in the Supplementary Information (task: Supplementary Methods II, organizations: Supplementary Methods III, post-task questionnaires: Supplementary Methods IV).

## Data analysis

Data was analyzed with custom code on MATLAB R2020b, R 3.4.1, and Python 3.11.5. Data met the assumptions of the statistical tests. All data analysis procedures and sample size determination procedures have been transparently reported in this manuscript. Detailed checks and validations are provided in the Supplementary Information (sample size and data quality assessment): Methods V, behavioral analyses: Supplementary Methods VI.

We computed power for first-wave (N = 79) data and simulated power for sample sizes of up to 250 participants. We employed Mixed Linear Models (MLMs) of the confidence hypothesis, controlled for the veracity judgment and the interaction of news veracity with news theme. With $\alpha = 0.05$, the observed fixed effect of confidence on information-seeking choices ($\beta = -0.15$) replicates findings from the literature on confidence-based information-seeking[24] and yields a power = 0.99. For an estimated fixed effect twice lower ($\beta = -0.72$), a simulated N = 150 approximates a power = 0.99. For an estimated fixed effect three times lower ($\beta = -0.48$), a simulated N = 200 approximates a power = 0.99. We fixed a total sample size of N = 250 to adequately test study hypotheses and included 5–10% additional participants to account for potential outliers and dropouts (Supplementary Methods V.2).

After collecting data from the second wave, Bayesian analyses were conducted, modeling responses using beta-binomial or normal distributions with non-informative Jeffreys priors. Participant behavior consistency across groups and sessions was confirmed, leading to data pooling (Supplementary Methods V.3 and Supplementary Methods V.4).

To control for objective performance accuracy in veracity judgments, we compared the success proportion in estimating veracity against a random distribution using a logistic function within a Bayesian framework (Supplementary Methods VI.1). Our null hypothesis assumes a distribution of behaviors equivalent to randomness. We tested the probability of success at $p = 0.5$ and computed a Bayes factor to compare $p = 0.5$ and not $p = 0.5$. We defined a logistic function with priors for $lambda = 0.5$ and $rscale = 0.5$, iterating 10,000 times. Although this rscale is considered a medium value, it represents a tight distribution around the mean in our case. We also computed a logistic function with $rscale = 1.5$ for a wider distribution.

We tested our hypotheses of participants' behavior using repeated measures MLMs. We modeled success in estimating veracity (correct or incorrect), veracity judgment (true or false), confidence (level per trial), demand for more information (choice to receive or not), and willingness-to-pay (ECUs amount per trial). The random structure of our MLMs included random effects for participants. Registering to the experiment required respecting our inclusion criteria. However, we failed to make reporting age, sex, and education mandatory when fulfilling the socio-demographics fields at the beginning of the experiment. In total, three participants did not report their age, four did not report their sex, and 28 did not report their education. When accounting for the socio-demographics, we excluded 30 participants from the models. All models are available in Supplementary Methods VI (veracity estimation: Supplementary Methods VI.2, metacognitive abilities: Supplementary Methods VI.4, demand for further information: Supplementary Methods VI.5).

We also tested Bayesian hypotheses of success in estimating veracity through separate Bayesian multilevel linear models (see Table 1; see Supplementary Methods VI.3), aligning with models formulated for null hypothesis significance testing. Each model included the variables of interest, a simplified random structure (subject random effects) to save computation time, and weakly informative priors. Models were compared using information criteria, particularly the Widely Applicable Information Criterion (WAIC), which measures predictive accuracy for a new dataset and penalizes models based on their parameter count. Bayesian stacking was employed to average Bayesian predictive distributions, with model weights derived from their information criteria performance, indicating their probability of being the best in terms of out-of-sample prediction[57].

Finally, we ran a multiple moderated mediation model (see Supplementary Methods VI.6). We used a single model using bootstrapping to evaluate the significance of indirect effects across varying levels of the mediator and moderators. News content imprecision and propensity to polarize were the predictor variables, with veracity judgment moderating and confidence mediating their effects. Reception choice was the outcome variable. Confirmatory factor analysis ensured measurement adequacy and all factor loadings except news content propensity to polarize exceeded 0.6, while composite reliability and average variance extracted surpassed recommended thresholds (0.7 and 0.5, respectively)[58].

All analyzed data and questionnaires have been documented either in the manuscript or in the Supplementary Information.

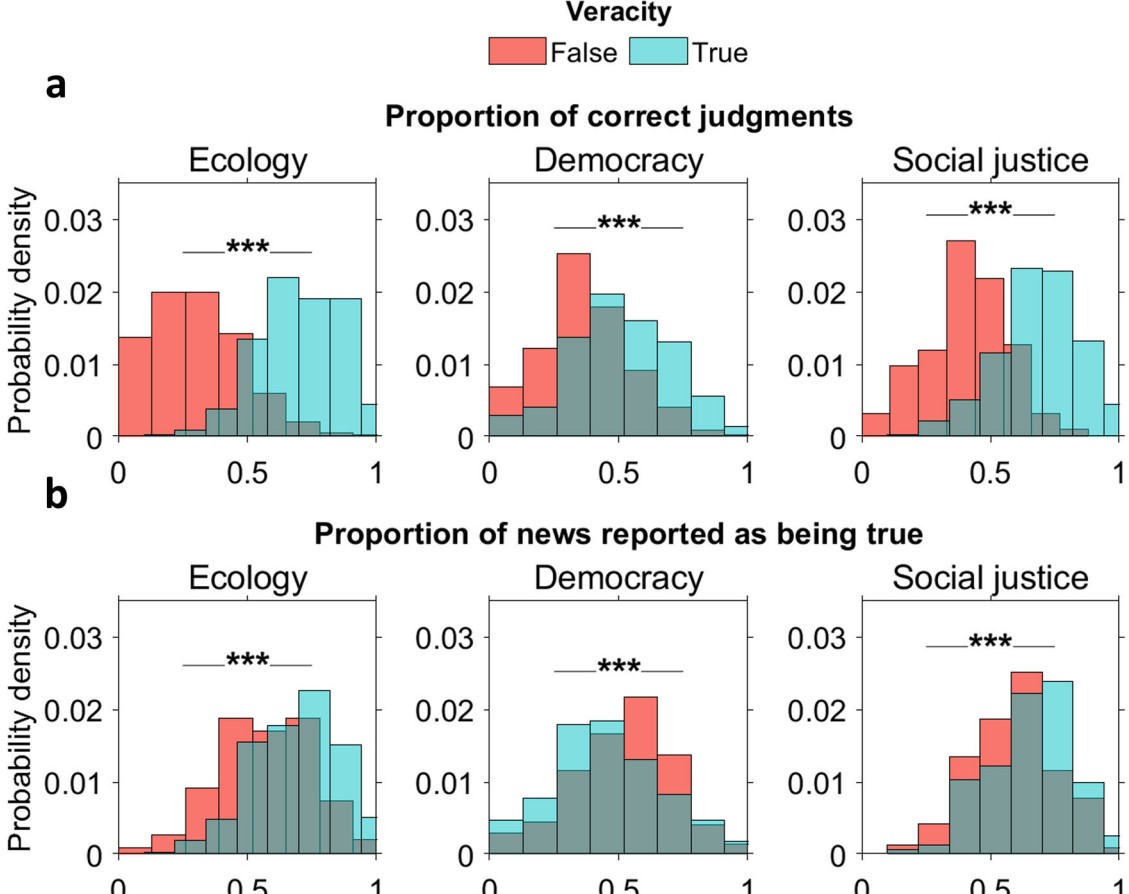

**Fig. 2 | Distributions of success and of veracity judgments. a** Probability densities of correct veracity judgment (i.e., the proportion of false news judged as false and of true news judged as true) are displayed separated by news themes (ecology, democracy, social justice) and news veracity (true or false). Individuals were better at evaluating news that was true than a news that was false. The likelihood of success was higher for news that was actually true. **b** Probability densities of news reported as being true are displayed separated by news themes (ecology, democracy, social justice) and news veracity (true or false). There was more news judged as true than false (i.e., Probability Density function skewed to the right), reflecting a bias toward judging news as true, with the exception of democracy-related news. n = 258.

## Reporting summary

Further information on research design is available in the Nature Portfolio Reporting Summary linked to this article.

## Results

### Judgment of news veracity and success rate

We first confirmed that performance accuracy was at a chance level when judging the veracity of the news. Participants' average success (i.e., correctly judging true news as true and correctly judging false news as false) rate was 51.6% (SD = 6.7)% (Supplementary Methods VI.1, Supplementary Fig. 2). A comparison of the performances with a random distribution within a Bayesian framework confirmed that performances were at chance level. Modeling random responses with a logistic function $\lambda \sim logistic(\lambda_0, rscale)$ with priors $\lambda_0 = 0.5$ and rscale = 1.5, the Bayes Factor (BF) favored the null hypothesis of chance level by a factor of about 0.3. This factor is considered the low boundary of moderate evidence[59], however, posteriors probabilities fell within the range of [49–53]% success, centered around 51.5%. (Supplementary Methods VI.1, Supplementary Figs. 3 and 4 & Supplementary Tables 1 and 2).

Next, we examined under what conditions participants' successes deviated from chance level. Although judgment of ambiguous news veracity was equivalent to chance, participants performed better with true news (M = 64%, SD = 11.9%) than with false news (M = 39.1, SD = 12%). The lowest accuracy was for democracy-related news (48.6 ± 11.5), with a slightly higher accuracy for news related to ecology (M = 52.2%, SD = 11.3%) and social justice (M = 53.8%, SD = 11.8%) (see Supplementary

Methods VI.2, Supplementary Table 3). Binomial Mixed Linear Models (MLMs) showed that participants predicted true news significantly more accurately than false news (odds-ratio = 2.83, 95% CI [2.62, 3.07], $p < 0.001$), with the veracity of news interacting with its theme ($p < 0.001$). Democracy-related news had significantly lowest accuracy compared to ecology and social justice (odds-ratio respectively at 1.19 and 1.26, 95% CI [1.06, 1.33] and [1.12, 1.41], $p = 0.0013$ and $p < 0.001$) (see Fig. 2a). Effects remained highly significant ($p < 0.005$) after controlling for socio-demographics, veracity judgment, and confidence (Supplementary Methods VI.2, Supplementary Table 4).

Such relatively higher ability to assess true news accurately can be explained by a general tendency to declare information as true (M = 59.5%, SD = 10.6%), with slightly more true news declared as true (M = 60.9, SD = 12.7%) than false news (M = 58.2%, SD = 3.1%) (see Supplementary Methods VI.2, Supplementary Table 5). Analyses supported that the veracity judgment predicted veracity perception (p = 0.003), withstanding the inclusion of control variables (Supplementary Methods VI.2, Supplementary Table 4). Interestingly, binomial MLMs of veracity judgment (Supplementary Methods VI.2, Supplementary Table 6) revealed that participants were especially more likely to judge as true ecology-related (prob. = 0.678, SD = 0.02) and social justice-related news (prob. = 0.637, SD = 0.02) than democracy-related news (odds-ratios respectively 1.71, 95% CI [1.52, 1.93], $p < 0.001$, and 1.43, 95% CI [1.27, 1.60], $p < 0.001$) (see Fig. 2b).

Results instead demonstrated that individuals' responses were primarily influenced by news ambiguity, specifically both the imprecision and

**Fig. 3 | Calibration analysis (i.e., degree of fit between a person's judgment of performance and his or her actual performance).** Participants' metacognition was not calibrated for estimating the probability of news veracity. The confidence-accuracy calibration plot displays the participants' accuracy in estimating probabilities that their judgment was correct as a function of their confidence level. Well-calibrated estimated probabilities would intersect with confidence degrees in the grey area, meaning, for example, that a 0–20% confidence degree predicts a 0–20% accuracy in evaluating news veracity. The plot shows that overall, the proportion of accurate veracity estimations did not increase nor decrease with confidence. Furthermore, the plot emphasizes that accuracy is higher for true news than for false news (the green curve always lies above the red one). Underconfidence dominates for true news whereas overconfidence dominates for false news. Note: n = 258.

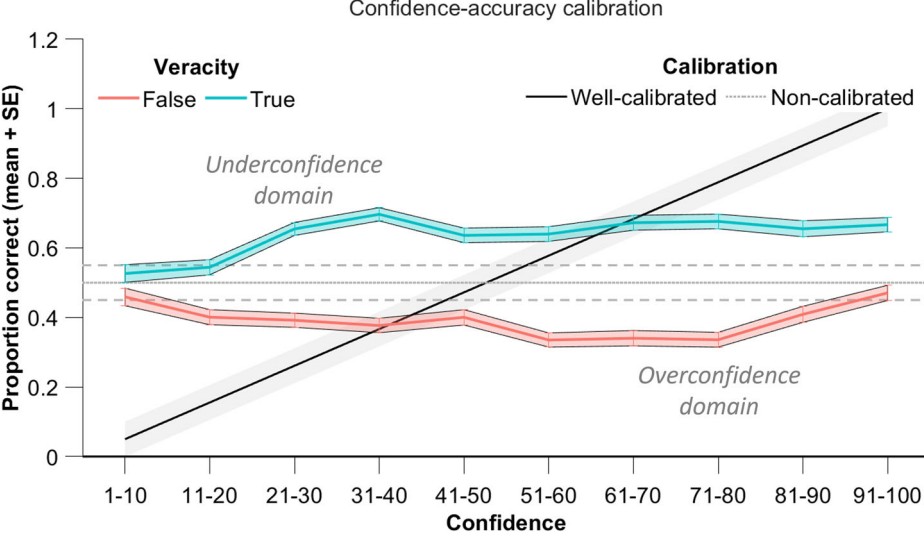

### Uncalibrated metacognitive sense of confidence

To investigate the relationship between confidence and accuracy in estimating veracity, we examined participants' calibration, that is their ability to accurately estimate the chances that the news is true or false (see Supplementary Methods VI.4, Supplementary Table 10). The confidence-accuracy calibration reflects, for given veracity judgments (the news is evaluated as true or false), the relationship between the continuous scale of confidence ([1,100]) and the binary outcome (true or false). This calibration indexes the extent to which confidence in one's judgment predicts the accuracy of this judgment. A perfect calibration is characterized by a linear confidence-accuracy function with 100% accuracy for 100% confidence, 90% accuracy for 90% confidence, etc. We sorted the individual confidence-accuracy relationships into ten bins and represented an area of well-calibrated estimation that spanned 10% (see Fig. 3).

We expected that participants' confidence would be non-calibrated and uncorrelated with actual success in estimating the veracity of ambiguous news. As the plot shows, participants' accuracy in estimating veracity was independent of their confidence in their estimation. Participants were neither well-calibrated nor ill-calibrated for estimating probabilities. Values above the diagonal signal underconfidence (individuals have a higher proportion of correct guesses than their reported level of confidence) while values below the diagonal reveal overconfidence (individuals have a lower proportion of correct guesses than their reported level of confidence). Figure 3 shows that underconfidence dominates for degrees of confidence below 50% whereas overconfidence dominates for degrees of confidence above 50%. Underconfidence dominates for true news whereas overconfidence dominates for false news, while news veracity judgment did not affect the relationships between confidence and success levels (Supplementary Methods VI.4, Supplementary Fig. 11). Bayesian mixed linear modeling supported the absence of a meaningful relationship between Confidence and Success. Although the model indicates a small positive coefficient for Confidence ($\beta = 0.05$, 95% CI [0.02, 0.09]), it explains only 0.12% of the variance in Success ($R^2 = 0.0012$, 95% CI [0.0002, 0.003]). With 100% of the posterior within the Region Of Practical Equivalence (ROPE) [−0.1, 0.1] and an interval null Bayes Factor $BF_{01} = 1/1830$, the null received the greatest support. The support interval [0.00, 0.10] reinforces this, as it remains entirely within the ROPE, indicating a negligible effect (Supplementary Methods VI.4, Supplementary Fig. 12).

To understand the determinants of confidence during the estimation of news veracity, we examined the sources of variability using MLMs of alignment of beliefs with concerns related to the news, socio-demographics, and response times. Moreover, we examined models predicting the effects of

the polarization of news content, potentially leading to a perception of falsity (Supplementary Methods VI.2, Supplementary Fig. 5). We modeled success in veracity judgment with MLMs incorporating imprecision and polarization predictors in interaction with news veracity (Supplementary Methods VI.2, Supplementary Table 7). Note that the news content imprecision and propensity to polarize (from 0 to 10) were obtained from ratings of a group of subjects (n = 55) independent from the actual participants in the experiment (see "Methods" section). The interaction effect of each predictor had a highly significant effect on the success of veracity judgment likelihood (both $p < 0.001$). Specifically, success in judging true news increased when their content imprecision and propensity to polarize were at their *minimum* (minimum/maximum, imprecision odds-ratio = 1.82, 95% CI [1.43, 2.32], $p < 0.001$; polarization odds-ratio = 2.17, 95% CI [1.59, 2.95], $p < 0.001$) Conversely, for false news, success increased with *maximal* imprecision (minimum/maximum odds-ratio = 0.53, 95% CI [0.41, 0.69], $p < 0.001$) and *maximal* propensity to polarize (minimum/maximum odds-ratio = 0.22, 95% CI [0.17, 0.30], $p < 0.001$). Furthermore, MLMs of veracity judgments showed that the likelihood of judging news as true decreased with increased imprecision (odds-ratio = 0.41, 95% CI [0.34, 0.49], $p < 0.001$) and the propensity to polarize (odds-ratio = 0.23, 95% CI [0.19, 0.29], $p < 0.001$). The effects in all models withstood the inclusion of socio-demographics, veracity-theme interaction, and confidence.

Finally, we found that alignment of beliefs with news concerns, distrust in experts, and socio-demographics had no significant effect on the accuracy of veracity judgments. Using MLMs (see "Methods" section; Supplementary Methods VI.2, Supplementary Fig. 6, Supplementary Tables 8 and 9), response times, likely reflecting a cognitive reflection, showed a positive effect on judgment accuracy (odds-ratio = 1.07, 95% CI [0.59, 1.56], $p = .007$), albeit not robust to the inclusion of other factors. We used Bayesian inference hypothesis testing to support these findings. Comparing Bayesian versions of the regression models (see "Methods" section, Supplementary Methods VI.3, Supplementary Fig. 7–10), the winning model featured interaction terms between news veracity and both news content imprecision and propensity to polarize (see Table 1). Overall, individuals' accuracy deviated from a chance level in reaction to variations in news ambiguity. Precision and apparent consensus about news content were interpreted as a signal of veracity, while imprecision and apparent polarization were seen as signals of falsity. Note that we found no significant difference between true and false news either in terms of imprecision (M == 5.53, SD = 1.24 vs. M = 5.17, SD = 1.25, *ranksum*, $p = 0.09$) or in terms of polarization (mean ± SD = 6.61 ± 1.62 vs. 6.22 ± 1.62, *ranksum*, $p = 0.3$).

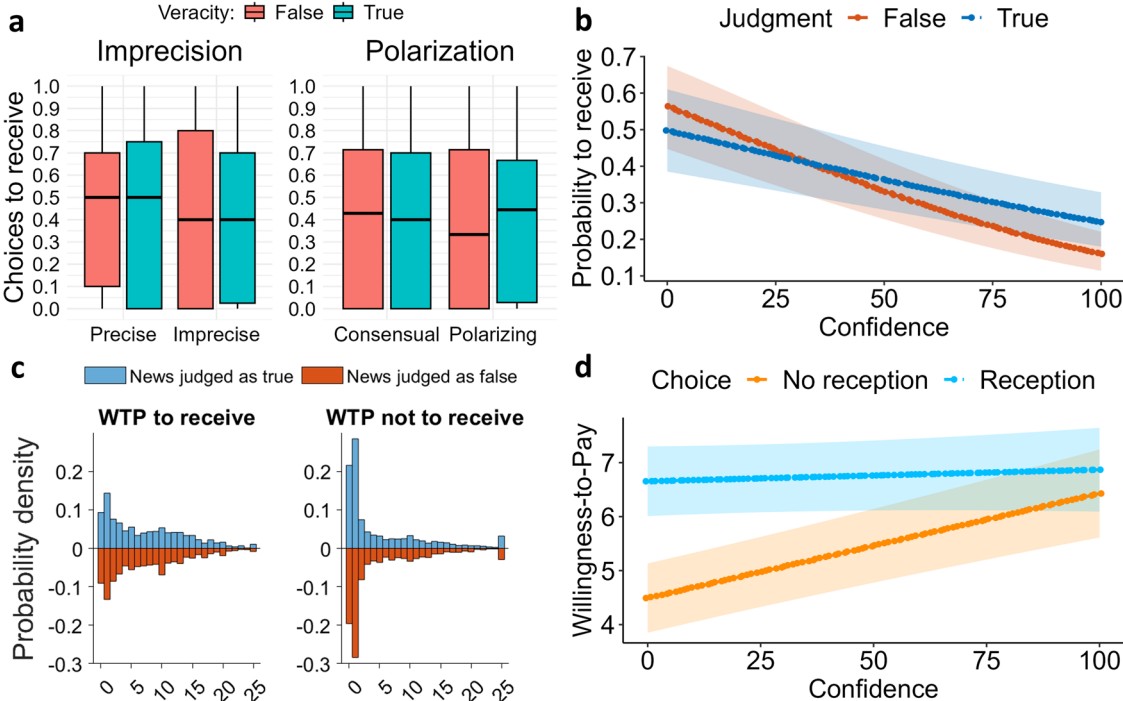

**Fig. 4 | The likelihood of choosing to receive extra information decreased as confidence in news veracity judgment increased. a** Panels show no decrease in the probability of acquiring extra information as imprecision or polarization increases (n = 258). **b** The probability of being willing to receive extra information about the news decreases as the confidence in one's judgment about news veracity increases. This decrease is steeper for news judged as false as compared to those judged as true. **c** The WTP (max: 25 EUC) was higher for the choices to receive extra information than for the choices to not receive it. **d** The WTP to receive extra information about the news was not affected by the degree of confidence in one's judgment about news veracity, whereas the WTP to avoid receiving extra information about the news increased with the degree of confidence in judgment about news veracity.

imprecision and polarization predictors on confidence in veracity judgments.

Alignment of beliefs with news concerns and socio-demographics showed no significant effects on confidence (Supplementary Methods VI.4, Supplementary Table 11), while the effects of response times were highly significant and negative ($p < 0.001$). Importantly, the interaction of both ambiguity predictors with the judgments of news as true decreased confidence ($p < 0.001$), even after including control variables (Supplementary Methods VI.4, Supplementary Table 12). Sex also revealed higher confidence levels for male participants than female participants ($p < 0.001$).

Confidence ratings were reliably affected by news content ambiguity, reflecting higher confidence that news is true under low ambiguity and higher confidence that news is false under high ambiguity. Comparing confidence levels between judgments of true and false news across three different levels of content imprecision and propensity to polarize revealed a significant effect of these variables on confidence. Even after the inclusion of control variables, confidence was higher for judgments of the news as *true* when imprecision was lowest (t ratio = 3.85, $p < 0.001$) and median (t ratio = 2.60, $p = 0.0092$). In contrast, confidence was not significantly different for judgments of the news as *false* than for judgments of the news as true when imprecision was at its highest level (t ratio = −1.84, $p = 0.065$). Conversely, confidence was higher for judgments of the news as *true* when the news content propensity to polarize was at its lowest level (z ratio = 8.61, $p < 0.001$), but higher for judgment of the news as *false* when polarization was highest (z ratio = −8.34, $p < 0.001$). These findings support the use of imprecision and polarization as signals of falsity, influencing veracity estimation.

**Demand and avoidance of extra information**

Next, we analyzed the demand for or avoidance of extra information about news that might resolve ambiguity. We predicted that despite the lack of calibration (i.e., a low degree of fit between confidence in news veracity judgment and the actual accuracy), individuals would use their metacognitive sense of confidence to decide whether or not to demand extra information about the news. Hence, we expected confidence to primarily explain the demand for extra information, particularly when confidence was low. First, we present participants' reception choices and subsequent willingness-to-pay (WTP). Then, we explore linear relationships between confidence and reception choices/WTP. To test our hypothesis, we estimated separate MLMs with variables capturing the main and interaction effects of participant confidence and news veracity judgment. The dependent variables were the binomial demand for more information or the continuous WTP. Post-hoc comparisons were conducted on estimated marginal means.

82.9% of participants demanded extra information at least once, with an average frequency of 42.29% (SD = 31.9%). Choice of extra information did not significantly differ between news themes (Kruskal Wallis Chi square = 4.39, $p = 0.11$, df = 2; democracy: M = 41.04%, SD = 33.16%; ecology: M = 43.27%, SD = 33.67%; social justice: M = 42.56%, SD = 33.09%; see Supplementary Methods VI.5, Supplementary Table 13). Participants chose to receive extra information 42.51% (SD = 32.44%) of the time when the news was judged as false and 42.07% (SD = 32.14%) of the time when the news was judged as true. Bayesian modeling of reception choices between judgments (Jeffreys priors: α = 0.5, β = 0.5) revealed a negligible difference (delta = 0.23, 95% Credible Interval [−0.008, 0.012]), indicating similar demand for extra information regardless of veracity judgments.

Participants exhibited a higher willingness-to-pay (WTP) for receiving extra information (M = 7.07 ECU, SD = 4.96 ECU) compared to not receiving it (M = 5.75 ECU, SD = 5.69 ECU) (see Fig. 4; see Supplementary Methods VI.5, Supplementary Table 14). Bayesian models of WTP for receiving and not receiving extra information (Jeffreys priors: μ = 0, σ = 1 from half-Cauchy distribution) showed that participants were willing to pay

**Table 2 | The effect of news content imprecision and polarization, moderated by the veracity judgment, on the reception choices, is mediated by the confidence in the veracity judgment**

| Moderated mediation model | | | | | | | | |
|---|---|---|---|---|---|---|---|---|
| Outcome | Predictor | SE | Z | p | b | 95% CI (b) | b* | 95% CI (b*) |
| Reception | Imprecision | 0.01 | −1.79 | 0.073 | −0.03 | [−0.06, 0.00] | −0.03 | [−0.07, 0.00] |
| Reception | Polarization | 0.01 | 0.23 | 0.23 | 0 | [−0.02, 0.03] | 0 | [−0.03, 0.04] |
| Reception | Confidence | 0 | −13.96 | <0.001 *** | −0.01 | [−0.01, −0.00] | −0.15 | [−0.18, −0.13] |
| Reception | Judgment | 0.11 | −0.43 | 0.666 | −0.05 | [−0.26, 0.16] | −0.02 | [−0.13, 0.08] |
| Reception | Imprecision x Judgment | 0.02 | 0.93 | 0.353 | 0.02 | [−0.02, 0.06] | 0.05 | [−0.05, 0.15] |
| Reception | Polarization x Judgment | 0.02 | 0.38 | 0.707 | 0.01 | [−0.03, 0.04] | 0.02 | [−0.08, 0.12] |
| Confidence | Imprecision | 0.32 | −3.93 | <0.001 *** | −1.25 | [−1.89, −0.62] | −0.06 | [−0.09, −0.03] |
| Confidence | Polarization | 0.3 | 6.32 | <0.001 *** | 1.87 | [1.31, 2.44] | 0.11 | [0.07, 0.14] |
| Confidence | Judgment | 2.56 | 6.21 | <0.001 *** | 15.92 | [10.87, 20.92] | 0.28 | [0.19, 0.37] |
| Confidence | Imprecision x Judgment | 0.43 | 2.43 | 0.015 * | 1.05 | [0.20, 1.95] | 0.1 | [0.02, 0.19] |
| Confidence | Polarization x Judgment | 0.37 | −8.34 | <0.001 *** | −3.08 | [−3.81, −2.36] | −0.36 | [−0.45, −0.28] |

The table displays a moderated mediation model. Confidence is the single variable with a direct effect on the reception choices. Crucially, confidence mediates the effect of news content imprecision and news content propensity to polarize, conditional on the veracity judgment, on the reception choices. The model is described in "Data analysis" section of the "Methods" section. b = unstandardized coefficients. b* = standardized coefficients.
CI confidence interval.
*p < .05; ***p < .001.

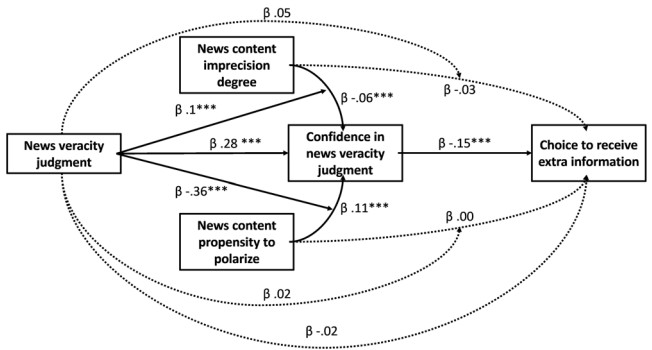

**Fig. 5 | Mediation effect of confidence, moderated by news content imprecision and propensity to polarize, predict the demand for extra information.** News content imprecision and news content propensity to polarize, conditional on the veracity judgment, have indirect effects on reception choices (*i.e.*, the decision to acquire extra information about the news) via the confidence in the veracity evaluation. Indirect effects are represented with dotted lines; direct effects are represented with solid lines. The coefficients are standardized. Notes: *p < 0.05.
**p < 0.01. ***p < 0.001.

more to receive it than to avoid it (delta = 1.327, 95% Credible Interval [−2.302, −0.344]).

As predicted, confidence explains the demand for extra information (p < 0.001, odds-ratio = 0.25, 95% CI [0.17, 0.37], *p < 0.001*), with a significant negative interaction with veracity judgment (p < 0.001). These effects remained significant even after incorporating controls such as the interaction of news veracity and theme and socio-demographics (Supplementary Methods VI.5, Supplementary Table 15). The results show that the probability of demanding extra information is not affected by news content ambiguity (i.e., imprecision and propensity to polarize) (see Fig. 4a) while it decreases as confidence in one's judgment increases. Specifically, the decrease is more pronounced when the news is judged as false (minimum/maximum confidence; judgment as false, odds-ratio = 6.41, 95% CI [3.96, 10.36], *p < 0.001*; judgment as true, odds-ratio = 2.59, 95% CI [1.68, 3.97], *p < 0.001*) (see Fig. 4b). A regression analysis of WTP further supported these findings, revealing a significant interaction between confidence and the demand for information (p < 0.001) (see Fig. 4c), holding up against the inclusion of control variables (Supplementary Methods VI.5,

Supplementary Table 16). According to this model, the effect size of confidence (*minimum – maximum* confidence levels) on the WTP when participants opted not to receive extra information was −1.74 (p < 0.001), whereas the effect size for the WTP to receive extra information was only −0.13 (p = 0.69) (see Fig. 4d).

The alignment of beliefs with news concerns from only two organizations predicted reception choices while we found no evidence for the effects of socio-demographics, response times, distrust, or ambiguity on decisions to seek information that might resolve ambiguity about the news.

To sum up, there is a significant inverse relationship between the demand for extra information about the news and confidence in one's judgment about news veracity. Moreover, this relationship is stronger for the news that participants judged as false. Supporting these findings, participants are also willing to pay more to not receive more information about what they think they already know.

A moderated mediation analysis further extended the role of confidence in the estimation of ambiguous news veracity (Table 2, Fig. 5; Supplementary Methods VI.6, Supplementary Fig. 13). Confidence had a unique direct effect on the outcome reception choice (standardized interaction $\beta = -0.15$, Z = −13.96, p < 0.001). Its effect was specifically a mediator effect, whereby the ambiguity of news, that is, news content imprecision and news content propensity to polarize, had an indirect effect on the reception choices through the confidence (imprecision: standardized interaction $\beta = -0.06$, Z = −3.93, p < 0.001; polarization: standardized interaction $\beta = 0.11$, Z = 6.32, p < 0.001). Veracity judgment played a role by moderating the effect on the news content imprecision to confidence path (standardized interaction $\beta = 0.1$, Z = 2.43, p = 0.015) as well as the effect on the news content propensity to polarize to confidence path (standardized interaction $\beta = -0.36$, Z = −8.34, p < 0.001). This analysis shows that the uncalibrated metacognition operating during the evaluation of true and false news induces a demand for disambiguating information that is increasingly ineffective as individuals are lured by the ambiguity of the news.

## Discussion

Headlines in the real world often do not overtly appear true or false, but instead fall into an ambiguous gray area, which makes them more difficult to evaluate. Using a novel experimental design, we carefully selected non-partisan and non-ego-relevant news that offer various levels of content

imprecision and polarization. The study was designed to address the complexity of evaluating ambiguous news, particularly within the context of misinformation. By varying the ambiguity of objectively verified true and false headlines, we controlled for subjective biases and ensured a range of cognitive responses. This approach allowed us to study metacognitive processes in a non-partisan context, avoiding reliance on extreme, obvious headlines, or emotionally charged content. Although 93.75% of the stimuli were predominantly categorized as neutral by our sentiment analysis, we observed a slight skew in emotional valence (Supplementary Methods V.1, Supplementary Fig. 1). False information exhibited a broader range of negative sentiment ratings, while true information showed a wider range of positive ratings. This subtle difference aligns with literature suggesting that false information often evokes more negative emotions[1]. Emotions may promote belief in fake news[60]. Despite this, the overall sentiment distribution remained largely neutral, indicating that the skew did not impact the neutrality of the materials.

Participants' accuracy in assessing news veracity hovering at chance level confirmed that we manipulated news with ambiguous contents, thereby allowing us to disentangle the effects of confidence from the effects of objective performance accuracy. We focused on news about ecology, democracy, and social justice whose utility was mainly cognitive[51]. That is, we chose news that could help individuals form more accurate beliefs about the state of the world, and that would neither threaten their identity nor affect their perception of how others would see them. The sentiment analysis confirmed the neutrality of the stimuli' emotional valence. The reason was to restrict as much as possible distortions in the demand for extra information that would result from motivated reasoning to protect one's image or identity.

How do individuals judge the veracity of ambiguous news? Participants systematically overestimated the prevalence of true news for headlines related to ecology and social justice (but not for news headlines related to democracy). This inclination, also known as truth bias[61,62], could stem from the automatic acceptance of statements and the cognitive strain associated with reevaluating previously acknowledged information[63]. It may also be that individuals are inclined to regard information as correct if it is deemed "good enough", avoiding a costly in-depth analysis[64,65]. An alternative perspective suggests that evolution has shaped human communication towards truthfulness, with altruism and trust as norms to ensure cooperation[66]. For instance, children tend to initially trust social partners[67]. Moreover, some defend that there is a prevailing inclination toward intuitive honesty among humans[68], leading individuals to anticipate a higher frequency of true statements in the information they encounter. It may also be that participants held a baseline assumption that information is true, given the prevalence of true information people encounter[69–71]. The absence of overestimation of the prevalence of true news for headlines related to democracy challenges a general explanation, and instead suggests that individuals have become more suspicious about the veracity of political news, or that congruence vs. incongruence between personal preferences and the political message conveyed in the headlines (which we did not measure) may influence veracity judgments.

While truthful communication is essential, signals must also convey useful information in the presence of ambiguity. Epistemic vigilance[72] has been proposed as an evolutionary tool, encouraging individuals to critically assess the veracity of statements. Our study reveals that participants consider ambiguity dimensions like content imprecision and polarizing tendencies. Higher imprecision and propensity to polarize increased the likelihood of individuals mistakenly declaring news as false with confidence. Caution should be exerted when drawing conclusions from the measures of ambiguity. Intra-class correlation coefficients (ICC) values indicated moderate reliability of raters on the scoring of news content imprecision and good reliability for the propensity to polarize. However, our results are consistent with previous research showing that individuals disproportionately prefer information that would provide a sense of certainty[73]. The imprecision in information content may signal unreliability, as it provides less clarity in the verifiability of the assertion whereas in the face of conflicting information, content polarization may signal untrustworthiness. Ambiguous content could hinder coordination and impose cognitive strains, leading individuals to preferentially identify such content and avoid it as an epistemic strategy for truth-seeking. The prominence of these dimensions, especially in comparison to alignment with beliefs or distrust toward experts, is consistent with the fact that we manipulated news with a primary emphasis on cognitive utility.

Bayesian analysis revealed strong support for the hypothesis that participants' confidence did not reliably predict their accuracy in this task and we observed that their confidence-accuracy calibration was flat (Fig. 4). The news stimuli have been chosen to ensure performance at chance on average, with news that ranged in evaluation difficulty between very easy and very hard. Confidence usually strongly correlates with objective accuracy in perceptual decision tasks or adaptive behavior[24,28,29]. However, the relationship between one's accuracy of judgment and one's confidence about judgment is known to vary greatly with task difficulty, whereby confidence is decreasing predicting accuracy as difficulty increases[74–77]. This could be the case when the false bit of news is not the central idea but a peripheral idea of the news. The dissociation that we observed between confidence and actual success rate suggests a pattern specific to ambiguous news, in contrast with perceptual information, with individuals struggling to gauge their level of knowledge when confronted with potential misinformation. In tasks with perceptual information, the state of the world is directly accessible and potentially identifiable with enough time to accumulate evidence. In the case of textual information, prior knowledge could theoretically aid in the evaluation of stimuli. However, active engagement and motivation to consider prior knowledge are necessary for effective evaluation[5]. Such processes do not occur routinely during the comprehension of textual stimuli. A key factor that may influence whether individuals engage in careful evaluation is their beliefs about their own susceptibility to misinformation, pointing to an additional metacognitive dimension[78].

Crucially, although individuals held an inaccurate perception of their own knowledge, this metacognitive sense of confidence was the most decisive dimension that guided information-seeking behavior in our experiment. Participants were willing to pay more to not receive more information about news that they estimated they already knew to be false. These results suggest that the decision to seek additional information likely stems from the expected benefit of this additional information in terms of subsequent cognition and reduction of ambiguity about the state of the world. This key finding presumably reflects that individuals use ambiguity – reflected in their confidence in their judgment – to choose whether to gather more evidence[24–27,37,79]. To fully contextualize our findings, it is important to acknowledge a limitation of the design. Participants could choose to ignore the post-task email or not read the additional information it might contain, rather than paying to avoid receiving additional information. This behavior could not be controlled and stems from the manipulation of written news media. If the reception of information had been endogenous to the task, verifying that it was read would have required conditioning rewards on responses to questions about the news. However, this would have shifted the utility of the information from cognitive to instrumental, which would be adversarial to testing our hypotheses.

The present study provides empirical evidence indicating the challenges individuals face in distinguishing true from false ambiguous news, often confusing precise or consensual information with the truth. Our novel findings underscore the prime role of metacognitive abilities in mediating the relationship between ambiguous information assessment and the demand or avoidance of extra information. Individuals misjudge what they know but they also seek to receive information according to what they know. As a consequence, they misidentify shortfalls in their knowledge, preventing them from filling the gaps. Individuals lacking awareness of their susceptibility to inaccurate information may fail to engage in the correct evaluation strategies[78]. This demonstrates that individuals are not only at risk of receiving undetected false information but also inefficiently exploring their environment, potentially spreading false information upon sharing it[35].

While previous literature suggests that people share false information due to a lack of attention to accuracy[16,17], our study suggests that their search for information to reduce ambiguity is driven by misplaced confidence in their veracity judgment. The structural equation modeling suggests that this search is increasingly ineffective as individuals are lured by the ambiguity of news. These findings are all the more important as our societies are facing major challenges with the extremely fast technical development of generative AI and the spread of deepfakes that will make the identification of veracity more and more difficult in the immediate future. These findings demonstrating the importance of metacognition in the assessment of the veracity of ambiguous news and in the search for information is very consistent with recent research showing a pivotal role of metacognition in belief updating in sensitive domains, such as in politically contested domains[80].

Our results highlight potential interventions and modifications to social media features that complement existing approaches for addressing misinformation and detecting truth. They call for testing within education and media literacy programs[81] approaches targeting individuals' ability to estimate veracity and to engage in self-motivated extra information seeking[78]. This includes exploring methods that encourage people to estimate their confidence in news content and validate it against evidence to increase awareness[22]. It includes as well training with specific search heuristics[11,82–85] and probability calibration exercises to help people improve their assessment of their own knowledge and their need for further information-seeking[83,84,86]. These interventions complement news content moderation, signaling of trustworthiness, and changes in the incentive structure of media platforms[12,13,87,88], aiming both to decrease motivations to share content that receives high social reward at the cost of accuracy and to increase accuracy motivation[17,36].

## Conclusions and limitations

To conclude, our study reveals that: (1) accuracy, i.e., correct veracity judgment remains consistent across topics, with a higher proportion of correct judgments for true versus false information regardless of the topics of the news (ecology, democracy, social justice); (2) confidence did not reliably predict accuracy in this task, with confidence-accuracy calibration remaining flat across confidence levels; (3) news imprecision and news tendency to polarize opinions increase the likelihood of confidently misinterpreting both true and fake news; (4) the topics of news headlines influence the proportion of news reported as being true (true-news bias) or false (false-news bias). That is, there is a true-news bias for news related to ecology and social justice while there is a false-news bias for news related to democracy; (5) confidence in one's judgment drives the willingness to seek additional information to better assess news veracity. Note that the distinction between judgment accuracy and response bias is important because most studies on misinformation focused on accuracy, which conflates the ability to correctly distinguish between true and false ambiguous news with the general tendency to judge news as true or false[89].Several limitations warrant consideration. While we controlled for headline topics and conducted sentiment analysis, a more granular examination of headline characteristics could further illuminate veracity judgments. Additionally, our focus on headlines alone, excluding other news media formats (e.g., full-length articles, audio-visual information), may limit generalizability. Finally, while our study centered on how confidence predicts veracity judgment and information-seeking behavior, we did not extensively investigate other psychological factors affecting misinformation susceptibility. In line with this question, a recent study reported that U.S. participants with higher analytical thinking skills have a greater ability to differentiate true from false news (here called proportion of correct judgments). In contrast, true-news bias (the tendency to label news as true) was linked with ideological congruency (alignment of participants' political identity with the political lean of news headlines), motivated reflection, and self-reported familiarity with news[89]. Future theories of information seeking will need to integrate news characteristics, sociodemographic factors, and psychological factors—including metacognitive abilities—to comprehensively understand how individuals evaluate information veracity.

## Data availability

All raw and processed data used for the main analyses and supplementary information are freely accessible in .csv format via OSF: https://osf.io/436pq/.

## Code availability

The custom code (https://doi.org/10.5281/zenodo.14111549) used to produce the results are freely accessible format via OSF: https://osf.io/436pq/. All analyses were carried out with MATLAB version R2020b, Python version 3.11.5, and R version 4.3.1.

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

## Acknowledgements
This research has benefited from the financial support of IDEXLYON from Université de Lyon (project INDEPTH) within the Programme Investissements d'Avenir (ANR-16-IDEX-0005) and of the LABEX CORTEX (ANR-11-LABX-0042) of Université de Lyon, within the program Investissements d'Avenir (ANR-11-IDEX-007) operated by the French National Research Agency. This work was also supported by grants from the Agence Nationale pour la Recherche to J.C.D. (ANR-21-CE37-0032 and ANR-24-CE37-4261), and by MITI-2020 CNRS (MITI-2020-247719) to JCD and M.C.V. The funders had no role in study design, data collection, and analysis, decision to publish or preparation of the manuscript. We thank Pr Edmund Derrington for critically reading and correcting English in the draft of the manuscript.

## Author contributions
CRediT author statement: Valentin Guigon: Conceptualization, Methodology, Software, Validation, Formal analysis, Investigation, Data curation, Writing – original draft, Writing – review & editing, Visualization. Marie Claire Villeval: Conceptualization, Methodology, Validation, Resources, Writing – original draft, Writing – review & editing, Supervision, Project administration, Funding acquisition. Jean-Claude Dreher: Conceptualization, Methodology, Validation, Resources, Writing – original draft, Writing – review & editing, Supervision, Project administration, Funding acquisition.

## Competing interests
The authors declare no competing interests.
