## [Peer review file · Communications Psychology]

Metacognition Biases Information Seeking in Assessing Ambiguous News

Corresponding Author: Dr Jean-Claude Dreher

Version 0:

Decision Letter:

Dear Jean-Claude,

Thank you for your patience during the peer-review process. Your manuscript titled "Metacognition during fake news detection induces an ineffective demand for disambiguating information" has now been seen by 3 reviewers, and I include their comments at the end of this message. They find your work of interest but raised some important points. We are interested in the possibility of publishing your study in Communications Psychology, but would like to consider your responses to these concerns and assess a revised manuscript before we make a final decision on publication.

We therefore invite you to revise and resubmit your manuscript, along with a point-by-point response to the reviewers. Please highlight all changes in the manuscript text file.

Editorially, we consider the reviewers' concerns about confounds or lack of controls most significant. For example, two reviewers call into question the role of the stimulus material in the finding that participants' confidence in their judgments of news veracity is not related to accuracy and there are repeated concerns regarding the incentivisation of choices. These issues need to be addressed through further empirical work, which may also take the form of further analysis or analysis of other (existing or novel) data.

You will also need to improve the clarity of reporting and integration of the present study into the literature. As you revise the text for greater clarity, please take note of the order of sections in Communications Psychology - the Methods section appears before the Results.

I am attaching an Editorial Requests Table that details critical reporting requirements for the revised manuscript. Please attend to each item and ensure your manuscript is fully compliant. We are requesting that your manuscript aligns with these requirements as this facilitates the evaluation of your manuscript, reducing delays in re-review and potential future acceptance. If your revised manuscript is not aligned with these requests on major issues, such as those concerning statistics, it may be returned to you for further revisions without re-review. Additional information can be found in our style and formatting guide Communications Psychology formatting guide.

Please use the following link to submit your

- revised manuscript,
- point-by-point response to the referees' comments,
- cover letter (as a separate document),
- the Editorial Policy Checklist (see below),
- the Reporting Summary (see below), and
- the completed Editorial Request Table (attached):

Link Redacted

Best regards,

Marika

Marika Schiffer, PhD
Chief Editor
Communications Psychology

REVIEWER EXPERTISE:

Reviewer #1: misinformation

Reviewer #2: misinformation, decision making

Reviewer #2: misinformation, Bayesian reasoning

REVIEWER REPORTS:

Reviewer #1 (Remarks to the Author):

This manuscript describes an experiment that examined confidence in accuracy judgments of news and how that confidence relates to demanding additional information. Participants judged the veracity of headlines on a scale from -100 to 100, which captured confidence, and then had to decide if they would like additional information about that topic. The headlines were intentionally ambiguous and performance accuracy was near chance. Confidence did not significantly predict accuracy. Ambiguity was related to confidence judgments. Participants requested additional information more often when their confidence was low.

I found this manuscript interesting and well written. These findings make a nice contribution to the (understudied) role of metacognition in judging the veracity of news stories. I have just a few comments that could be addressed in a revision.

1. I understand why the authors used ambiguous headlines, but I worry that methodological detail 1) creates a study with poor ecological validity, and 2) could mask positive correlations between confidence and accuracy. Using some headlines that appear true to people and others that appear false would better mimic real world headlines. Further, confidence and accuracy in those headlines might be greater than in the ambiguous headlines. I think this should be discussed.

2. The finding that participants tended to believe news is consistent with idea of a truth bias (<https://doi.org/10.1177/0261927X17744004>; also see review by Brashier & Marsh's, 2020, <https://doi.org/10.1146/annurev-psych-010419-050807>). I think this should be added to the discussion of that finding.

3. The findings about metacognition share some similarity with these papers: <https://doi.org/10.1037/xlm0000977>, <https://doi.org/10.1080/10810730.2021.1955312>. I think the authors should discuss these similarities.

Reviewer #2 (Remarks to the Author):

This paper examines participants veracity judgments and confidence-accuracy calibration when assessing ambiguous claims. Additionally, it examines participants' requests, and willingness to pay, for additional disambiguating information. The relationships between confidence and desire for disambiguating information are interesting and potentially informative for researchers, as are the potential role that other information characteristics (imprecision, polarization, desirability) may have in driving requests for additional information. However, I have several points that I believe should be addressed prior to publication. I have outlined my specific points in more detail below.

1) There are some highly relevant papers that are not cited or discussed within the manuscript. For example, Donovan and Rapp (2020) examined whether giving participants the opportunity to look up additional information reduced misperceptions.

Also, Salovich and Rapp (2021) examined people's estimated vs. actual susceptibility to misinformation. More broadly, there is also a solid body of research on the role of lateral reading for reducing misperceptions, some of which may be relevant to the current study (for a recent review see McGrew, 2024)

2) Reference 21 does not seem relevant to the sentence "An alternative approach could involve targetting individuals themselves and focusing on enhancing their abilities to assess the veracity estimation." Can the authors please select a more appropriate reference and/or clarify why this reference is relevant and appropriate. There is also a typo in "targetting".

3) Given the items selected for inclusion were specifically chosen to ensure that they were ambiguous and performance would be at or close to chance, it seems very unsurprising that confidence ratings were uncalibrated with accuracy? Is this lack of calibration not just a function of the stimuli chosen? To have decent confidence accuracy calibration across the confidence range, shouldn't there be variation not only in the confidence ratings participants can provide but also in their performance on the items? For example, if items are intentionally selected so that there are no items on which participants are 80% accurate, then should never select 80% confidence? Instead, participants should have just selected 50% confidence for all trials because they were performing close to chance? Can the authors please either explain why my concern is wrong (I am open to counterpoints or pushback) or update the manuscript to note this as a limitation and appropriately adjust their conclusions.

4) Regarding the confidence elicitation, I am also not entirely sure why the robots explanation and procedure is necessary? Those instructions seem unnecessarily complicated and unclear. Would it not have been sufficient, and easier for participants to understand, to simply tell them that, for each item, an answer with X chance of being correct will be selected. If their confidence is below the threshold, the answer will replace their own, whereas if their confidence is above the probability of the answer being correct, their answer will be kept. Can the authors please clarify why this was necessary and/or note it as a potential limitation.

5) Unless I have missed something, I am not entirely sure why participants would ever want to request additional information if they also wanted to maximize the money they received? I can see a reason to request additional information if that information was provided within the experiment and participants were then able to update their veracity judgments and improve their chance of being rewarded. However, receiving the information after the experiment seems to mean that any request for additional information will reduce the total money they receive? Couldn't participants therefore put the minimum willingness to pay and just search themselves about anything they found interesting or wanted to know more about? Similarly, although I understand the desire to balance the cost of receiving vs. not receiving information, it is even less clear to me why participants would ever pay money to not receive information? Surely receiving the information does not incur any cost even if they do not want it because they could simply choose to ignore the email or not read the information? As with the above points, can the authors please either clarify if I have misunderstood something or am wrong, or note this as a limitation and adjust their conclusions appropriately.

6) I have some concerns about the pretesting that was conducted to select items for inclusion. Specifically, the authors state that 55 independent raters assessed the 210 headlines, but only assessed 42 headlines each. That means that each headline was only rated by 11 raters, which seems to be too few to ensure that you have reliable ratings? The average random raters' correlation coefficients reported in Supplementary II also seem quite low (0.54 for content imprecision and 0.59 for desirability, consensuality seems decent at 0.81).

7) Additionally, is noted in Supplementary II that desirability was predictive of requesting more information. However, only imprecision and polarization were considered within the main body of the paper. Can the authors please either also report the impact of desirability and/or explain why this would be inappropriate.

8) The organization ratings seem to confound familiarity and closeness. It seems perfectly plausible that someone could be very familiar with an organization and strongly oppose them, which may make averaging inappropriate. Can the authors report the reliability for their combined measure and/or better justify combining them.

9) In the discussion (Page 9, paragraphs 2 and 3) it is noted that participants overestimated the amount of true news and several potential explanations are provided. One important potential explanation that should be considered is that the vast majority of information that most people encounter is true (Acerbi et al., 2022; Guess et al., 2019). Therefore, having a baseline assumption that information is true may also be rational based on prior experience. The data collected regarding the perceived percentage of fake news from the various sources may also provide some insight into this (i.e., if they perceive a small percentage of information coming from those sources to be fake news, and particularly for the overall question about the internet in general, that may help explain why they generally assumed information was true and overestimated).

10) Finally, I request that the authors add a statement to the paper confirming whether, for all experiments, they have reported all measures, conditions, data exclusions, and how they determined their sample sizes. The authors should, of course, add any additional text to ensure the statement is accurate. This is the standard reviewer disclosure request endorsed by the Center for Open Science [see <http://osf.io/hadz3>]. I include it in every review.

References mentioned in the review:

Acerbi, A., Altay, S., & Mercier, H. (2022). Fighting misinformation or fighting for information? HKS Misinformation Review. <https://doi.org/10.37016/mr-2020-87>
Donovan, A. M., & Rapp, D. N. (2020). Look it up: Online search reduces the problematic effects of exposures to inaccuracies. *Memory & Cognition*, 48(7), 1128–1145. <https://doi.org/10.3758/s13421-020-01047-z>

Guess, A., Nagler, J., & Tucker, J. (2019). Less than you think: Prevalence and predictors of fake news dissemination on Facebook. *Science Advances*, 5(1), eaau4586. <https://doi.org/10.1126/sciadv.aau4586>

McGrew, S. (2024). Teaching lateral reading: Interventions to help people read like fact checkers. *Current Opinion in Psychology*, 55, 101737. <https://doi.org/10.1016/j.copsyc.2023.101737>

Salovich, N. A., & Rapp, D. N. (2021). Misinformed and unaware? Metacognition and the influence of inaccurate information. *Journal of Experimental Psychology: Learning, Memory, and Cognition*, 47(4), 608–624. <https://doi.org/10.1037/xlm0000977>

Reviewer #3 (Remarks to the Author):

Review of “Metacognition during fake news detection induces an ineffective demand for disambiguating information” for *Nature Communications*.

The paper includes a study evaluating participants’ perception that a series of ambiguous and apolitical information is true or false and their confidence about it. The paper investigates the link between truth perception, confidence and willingness to learn more about the information.

While the empirical content might seem modest - a single study with a modest sample size, the study is quite ambitious and was developed with great care and attention (e.g., pilot study, pretest, attempts to control for a number of variables). For example, the news used span different domains and the sentiment/valence of the different pieces was assessed, along with people’s perception of key aspects of the news (e.g., potential to divide, ambiguity). So, well done for this.

However, because of the complexity of the study, the clarity of some of the points is hard to grasp – including important methodological choices – and their implications. For example, regarding the complex way to implement incentives (described on page 11 and 12) and the way to operationalise information seeking/avoidance. This lack of clarity together with a complex design also means that some of the analyses seem to be tentative and explorative rather than hypotheses driven (e.g., using values from a pre-test to predict participants’ behaviour).

I believe the paper can make a strong contribution to the field and I suggest below ways to clarify the methodological choices, and their implications, along with some other suggestions.

Method

You report running a sentiment analyses on the materials and report that those were mostly neutral – but a look at the distribution of sentiments show differences in the distribution of negative and positive emotions – the false one had a wider range of negative ratings and the real one had a wider range of positive emotion. Could this maybe be mentioned and addressed in the manuscript?

Could you explain why using the ratings or organisation as an assessment of World view and not a more classical measure? Also, you mention 12 organisations and 6 judgments – did people judge pairs of organisations? Or did they judge a randomly selected set?

I might have missed this bit of information, but I was not sure where the data was collected, whether it was done in the lab or from anywhere?

To measure confidence, why not asking participants about confidence? I can see how a probability question is information (what are the chances that this information is true vs. false) but a score ranging from -100 to 100 might seem a bit counterintuitive, because it is not a linear indicator of confidence (it is a reverse U shape).

I did not quite understand the reference to the karni procedure. The paper cited did not include evidence that incentives improved probability evaluation.

The text page 11 line 439 describes the involvement of robot that does not seem clear. What is the rationale for that? Why not saying something simpler: If your response is correct / better than that of an artificial intelligence?

For example, I was not sure what was meant by “if the randomly drawn robot had an accuracy level higher than their own reported degree of confidence, we would take the robot’s answer into account”. Sorry, this point is important but unclear.

Assuming my interpretation of the incentive system above is correct, do you think this set up could have incentivise people to give extreme ratings (meaning close to -100 or close to +100, thereby uncalibrating people’s judgment instead of making them more accurate? Why not rewarding based on the closeness to being correct? (e.g., a score of +100 for a real information = score of 50 ECU – score of +50 for a correct news = 25 ECU, with a penalty for scores that are wrong to incentivise against going extreme).

Could you explain how veracity was scored? Was it the case that any score between 1 and 100 for a true information was categorised as correct? Could the response scale with both veracity and 0-100 and false on the -100 to 0 side on the same single scale lead people to the perception that they answered about the extent to which the fact was true or false? Which would not quite be a measure of confidence? Could you also clarify how 0 was coded?

Could you explain why adding a willingness to pay measure of the information sought (or refused) – and not simply asking people if they wanted to read more about the issue? Did you think that a cost would change their answers?

For participants who did not want more information, if they did not “pay” enough for their decision, then, did you send people the information despite their choice? If that would be the case, it seems it would be an ethical issue that should be mentioned and discussed. This seems to be what is implied in the text: “had to report how much they were 449 willing to pay, between 0 and 25 ECU of their 200 ECU initial endowment, to have their decision 450 implemented (i.e., to receive or not receive further information), [...] If the WTP was lower than the price, no deduction was operated and the option the participant did not choose was implemented.”

Also, do you think that people’s greater willingness to pay to get information (vs. not to get it) could be motivated by the fact that they would not have to read it – providing a low willingness to pay avoid the risk of forgoing some of their endowment and they would simply delete the email?

The Supplementary materials include a lot of information. It would be useful for interested readers to have an index. I struggled for example to find some of the information referred to in the manuscript in the SM (e.g., I could not find Supplementary VI.2, about the comparison of outcomes across data collection waves. – I think it might actually be 3. In SM)

Page 2, line 56 – it is not clear what the characteristics that increase ambiguity are.

Could you provide a rationale for using response time? It is not clear what it is supposed to measure – e.g., intuitive processing?

Literature review/introduction/Discussion

There is quite a bit of research about confidence and belief quality in the literature – but those findings are described as focusing on “perceptual task”. I am not sure what that means. People are quizzed with factual questions to which they provide answers that are either correct or not and then, they are typically asked how many correct answers they provided overall. It is quite similar to what is being done in the article and could be better presented.

It might be helpful to give an example of news in the manuscript – as I think that the fact that the choice of choosing maximally ambiguous non-partisan content (for which you expected chance level veracity judgment) might also explain the lack of link between confidence and accuracy in veracity judgments. This is a potential boundary condition that could be mentioned in the discussion. It does not take away the value of the results – it simply limits the potential situations to which it could apply. The discussion does include a description of the neutrality of the news – but does not include the implication of this for the findings.

In the paper, uncertain is sometimes used instead of ambiguous. I am not sure this is the best terminology. For example, the sea level might rise 2 meters – is uncertainty information – but it is not what I would call ambiguous.

“We investigated the relationship between objective accuracy in judging news veracity and 105 confidence in this judgment, controlling for how news ambiguity influences the judgments of news 106 as true or false”. The research goal hints that ambiguity is manipulated and this is a bit misleading because at that stage people do not know whether it was the case, or how, but reading carefully the method would show that it is not manipulated. Mentioning that this was not a manipulation but that ambiguity was subjectively assessed by a separate sample (of 55 participants) would be useful. It would also be helpful to also provide the distribution of all the ratings taken in that pretest (imprecision, desirability [which is a measure of information seeking], consensuality/divisiveness and themes).

Partly related to this, I think quite a bit of space is devoted to the unpredicted interaction effect between confidence and ambiguity on information seeking. I do not put the results into question and I find the authors’ interpretation fair, but this result is given maybe too much credit and prominence in the abstract and in the discussion (e.g., in the abstract: “the level of confidence, although uncalibrated, was the primary driver of the demand for additional information about the news, with lower confidence driving a greater demand, regardless of its veracity judgment. This demand for disambiguating information, driven by the uncalibrated metacognition, was increasingly ineffective as individuals became more enticed by the ambiguity of the news.”). In the abstract, readers are unaware of the way this was measured and so might place too much weight on the value of that finding.

Why keeping the cost of seeking (or refusing) information the same? Information comes at a cost, at least an opportunity cost (i.e., time spent reading), or possibly money (e.g., buying newspaper) – whereas refusing information usually does not?

Could the authors provide a distribution of confidence ratings for true and false statements across different themes? (maybe in SM)? Is it possible that participants chose their responses randomly?

I do not see the analyses with response time and prior beliefs – the paper includes the following: “Alignment of beliefs with news concerns and socio-demographics showed no significant effects 216 on confidence (Supplementary I.3, Table S11), while effects of response times were highly 217 significant and negative ($p < 0.001$).” but Table S11 does not include response time measure.

The paper implies that people should seek more information about topics about which they feel less confident – but people could merely be driven by interest. It is a thought provoking question to consider whether people should spend time reading about topics that were chosen to be immaterial to important decisions.

Minor comments

- I found the mean +- SD formulation a bit confusing – (as a psychologist) – maybe it is common for other disciplines? I find that it seems to suggest that the mean only vary by the SD value, which is obviously not the case. I find the M (SD = XX) or (M = XX, SD = XX) better options.
- Maybe a slight rewrite would help to understand what is being predicted and what is used to predict it? “Modelling the success in estimating veracity confirmed that the veracity judgment was a highly significant explaining variable” or maybe here there is a typo and the statement should be “Modelling the success in estimating veracity confirmed that the veracity of the news was a highly significant explaining variable” – which could be made even clearer and simpler as “Analyses supported that the veracity of the news predicted veracity perception”?
- In SM these two statements seem to contradict each other: “Fifty-five independent raters (F=33, M=22; mean \pm SD age=26.2 \pm 4.78) received the 210 true and false news. Five groups of 11 French-speaking raters evaluated each a subset of 42 news out of the 210.”
- The correlation between the desirability of information (how much do you want to learn more) in the pretest and information seeking in the main study is described in the SM as weak but it seems could it is more medium ($R_{\rho}=0.349$).
- All odd ratios < 0.76 – I think you meant > 0.76
- Page 9, line 325 – gullibility seems a bit too pejorative? Maybe trust would be a better terminology?

EDITORIAL POLICIES

We ask that you ensure your manuscript complies with our editorial policies and reporting requirements.

To that end, we require revised manuscripts to be accompanied by two completed items: a reporting summary that collects information on study design and procedure, and an editorial policy checklist that verifies compliance with all required editorial policies.

- <https://www.nature.com/documents/nr-reporting-summary.zip>>Nature Research Reporting Summary
- <https://www.nature.com/documents/nr-editorial-policy-checklist.pdf>>Editorial Policy Checklist

All points on the policy checklist must be addressed. Your revised manuscript can only be sent back to the referees if these checklists are completed and uploaded with the revision.

Notes: If you have submitted a Stage 1 Registered Report, Review, Primer, Comment, or Perspective you do not need to submit these forms. If you have already submitted these forms, you may disregard this request.

Version 1:

Decision Letter:

Dear Dr Dreher,

Your manuscript titled "Metacognition Biases Information Seeking in Assessing Ambiguous News" has now been seen by our reviewers, whose comments appear below. In light of their advice I am delighted to say that we are happy, in principle, to publish a suitably revised version in Communications Psychology.

We therefore invite you to revise your paper one last time to address the list of editorial requests. At the same time we ask that you edit your manuscript to comply with our format requirements and to maximise the accessibility and therefore the impact of your work.

EDITORIAL REQUESTS:

The most significant remaining issue is a lack of clarity surrounding the statistical evidence for the inference that confidence does not predict participants' accuracy and that metacognitive abilities are uncorrelated with success in estimating news veracity. These statements claim the absence of an effect or difference, and the reporting must firmly link these to positive evidence for the absence of an association derived from Bayesian statistics or equivalence tests.

SUBMISSION INFORMATION:

OPEN ACCESS:

*** TRANSPARENT PEER REVIEW:** Communications Psychology uses a transparent peer review system. On author request, confidential information and data can be removed from the published reviewer reports and rebuttal letters prior to publication. If you are concerned about the release of confidential data, please let us know specifically what information you would like to have removed. Please note that we cannot incorporate redactions for any other reasons.

*** CODE AVAILABILITY:** All Communications Psychology manuscripts must include a section titled "Code Availability" at the end of the methods section. We require that the custom analysis code supporting your conclusions is made available in a publicly accessible repository at this stage; please choose a repository that generates a digital object identifier (DOI) for the code; the link to the repository and the DOI must be included in the Code Availability statement. Publication as Supplementary Information will not suffice.

*** DATA AVAILABILITY:**

Link Redacted

Best regards,

Marika Schiffer

Marika Schiffer, PhD
Chief Editor
Communications Psychology

REVIEWERS' COMMENTS:

Reviewer #1 (Remarks to the Author):

The authors have adequately addressed my previous concerns. I believe that this manuscript makes an interesting contribution to the literature. This study, like all studies, has some limitations but the authors have described them clearly and this work should motivate additional research.

Reviewer #2 (Remarks to the Author):

I would like to thank the authors for being so receptive to the comments of myself and the other reviewers. The authors have now provided sufficient additional detail and/or clarifications that my previous concerns have been addressed. I believe the revised manuscript is greatly improved and would make a strong contribution to the literature.

Reviewer #3 (Remarks to the Author):

I was already quite satisfied with the quality of the study and the paper before, and the authors have addressed my suggestions (and that of other reviewers) thoroughly (thank you!). I am now happy to recommend the manuscript for publication.

Responses to Reviewer #1

This manuscript describes an experiment that examined confidence in accuracy judgments of news and how that confidence relates to demanding additional information. Participants judged the veracity of headlines on a scale from -100 to 100, which captured confidence, and then had to decide if they would like additional information about that topic. The headlines were intentionally ambiguous and performance accuracy was near chance. Confidence did not significantly predict accuracy. Ambiguity was related to confidence judgments. Participants requested additional information more often when their confidence was low.

I found this manuscript interesting and well written. These findings make a nice contribution to the (understudied) role of metacognition in judging the veracity of news stories. I have just a few comments that could be addressed in a revision.

We are extremely grateful for your thorough report, which helped us a great deal in rewriting and refocusing our paper. We believe that the updated manuscript delivers a much sharper presentation of our core contribution and results. We thank you in advance for your patience in reassessing the new manuscript. In our responses below, we use bold font whenever we quote from your letter, and we use standard font for our responses, and italicized fonts for quoted text from the revised manuscript.

1. I understand why the authors used ambiguous headlines, but I worry that methodological detail 1) creates a study with poor ecological validity, and 2) could mask positive correlations between confidence and accuracy. Using some headlines that appear true to people and others that appear false would better mimic real world headlines. Further, confidence and accuracy in those headlines might be greater than in the ambiguous headlines. I think this should be discussed.

As noted in our introduction, our goal was to study misinformation, characterized by false, inaccurate or misleading information. Very often, headlines or news do not appear obviously false or true. In fact, many news are ambiguous (in the sense that individuals do not know the probability they are true), especially when they are vague or deal with novel topics, or come from non-official sources like in social media. Frequently people simply do not know whether these headlines/news are true or not. Therefore, we are not sure that ‘headlines that appear true to people and others that appear false’ would better mimic real world headlines.

Our study was precisely and purposely designed to investigate how individuals evaluate non-partisan ambiguous news. This also allowed us to extend the understanding of misinformation beyond partisan contexts, which has widely been studied in the literature (Bago et al., 2020; Pennycook, Epstein, et al., 2021; Pennycook & Rand, 2019). To achieve this, we varied the ambiguity of the headlines, as assessed by independent raters. We selected verified true and false news from debunking sources and modified some headlines to create a spectrum of ambiguity.

We think that our approach, utilizing objectively true or false news, helps control for subjective biases in participants' judgments. Headlines that merely *appear* true or false would introduce uncontrolled variations between individuals—what seems true to one person may seem false to another. We would lose control of this feature of the design and this would complicate the interpretation of results.

You are right that the lack of correlation between confidence and accuracy observed in our study could be attributed to the inherent variability in news ambiguity. Although our stimuli were objectively true or false, the varying levels of ambiguity introduced different levels of difficulty, offering a nuanced view of metacognitive processes. The flat confidence-accuracy calibration observed highlights the challenge of assessing ambiguous news rather than an issue with the headlines or their objective truth.

We addressed this point as follows page 13:

“The study was designed to address the complexity of evaluating ambiguous news, particularly within the context of misinformation. Headlines in the real world often do not overtly appear true or false, but instead fall into an ambiguous grey area, which makes them more difficult to evaluate. By varying the ambiguity of objectively verified true and false headlines, we controlled for subjective biases and ensured a range of cognitive responses. This approach allowed us to study metacognitive processes in a non-partisan context, avoiding reliance on extreme, obvious headlines.”

2. The finding that participants tended to believe news is consistent with idea of a truth bias (<https://doi.org/10.1177/0261927X17744004>; also see review by Brashier & Marsh’s, 2020, <https://doi.org/10.1146/annurev-psych-010419-050807>). I think this should be added to the discussion of that finding.

We thank you for these references that we have added to our discussion (see p.9):

“This inclination, also known as truth bias^{58,59}, could stem from the automatic acceptance of statements and the cognitive strain associated with reevaluating previously acknowledged information⁶⁰.”

3. The findings about metacognition share some similarity with these papers: <https://doi.org/10.1037/xlm0000977>, <https://doi.org/10.1080/10810730.2021.1955312>. I think the authors should discuss these similarities.

We have added to our discussion (see p.14.):

“In the case of textual information, prior knowledge could theoretically aid in the evaluation of stimuli. However, active engagement and motivation to consider prior knowledge are necessary for effective evaluation⁵. Such processes do not occur routinely during comprehension of textual stimuli. A key factor that may influence whether individuals engage in careful evaluation is their beliefs about their own susceptibility to misinformation, pointing to an additional metacognitive dimension⁷⁵.

[...]

Individuals lacking awareness of their susceptibility to inaccurate information may fail to engage the correct evaluation strategies⁷⁵.”

Once again, we thank you for your suggestions and comments and hope that we have satisfactorily addressed your points.

Responses to Reviewer #2

This paper examines participants veracity judgments and confidence-accuracy calibration when assessing ambiguous claims. Additionally, it examines participants' requests, and willingness to pay, for additional disambiguating information. The relationships between confidence and desire for disambiguating information are interesting and potentially informative for researchers, as are the potential role that other information characteristics (imprecision, polarization, desirability) may have in driving requests for additional information. However, I have several points that I believe should be addressed prior to publication. I have outlined my specific points in more detail below.

We are extremely grateful for your very careful reading of our paper and your thoughtful and constructive comments. We are also thankful for your clear guidance on how to revise our paper. In our responses below, we use bold font whenever we quote from your letter, and we use standard font for our responses, and italicized fonts for quoted text from the revised manuscript.

1. There are some highly relevant papers that are not cited or discussed within the manuscript. For example, Donovan and Rapp (2020) examined whether giving participants the opportunity to look up additional information reduced misperceptions. Also, Salovich and Rapp (2021) examined people's estimated vs. actual susceptibility to misinformation. More broadly, there is also a solid body of research on the role of lateral reading for reducing misperceptions, some of which may be relevant to the current study (for a recent review see McGrew, 2024)

We thank you for these references that we sorriily omitted and that we have now added to our discussion (see p.4 & p.15).

“A key factor that may influence whether individuals engage in careful evaluation is their beliefs about their own susceptibility to misinformation, pointing to an additional metacognitive dimension⁷⁵.

[...]

Individuals lacking awareness of their susceptibility to inaccurate information may fail to engage the correct evaluation strategies⁷⁵.

[...]

They call for testing within education and media literacy programs⁷⁷ approaches targeting individuals' ability to estimate veracity and to engage in self-motivated extra information seeking⁷⁵.

[...]

It includes as well training with specific search heuristics^{11,78-81} and probability calibration exercises to help people improving their assessment of their own knowledge and their need for further information-seeking^{79,80,82}.”

2. Reference 21 does not seem relevant to the sentence “An alternative approach could involve targetting individuals themselves and focusing on enhancing their abilities to assess the veracity estimation.” Can the authors please select a more appropriate reference and/or clarify why this reference is relevant and appropriate. There is also a typo in “targetting”.

We apologize for our mistake. Indeed, the article intended to be cited was not:

[List, J. A., Ramirez, L. M., Seither, J., Unda, J. & Vallejo, B. *Toward An Understanding of the Economics of Apologies: Evidence from a Large-Scale Natural Field Experiment. National Bureau of Economic Research* (2024). doi:10.1093/ej/ueab062].

It was instead:

[Lichtenstein, S., & Fischhoff, B. (1980). Training for calibration. *Organizational behavior and human performance*, 26(2), 149-171.]

We have fixed this issue as well as the typo on ‘targeting’.

We also added the following reference: (*Callender, Franco-Watkins & Roberts, 2016, Improving metacognition in the classroom through instruction, training, and feedback.*) in which the authors argue that training, as pairing multiple trials of confidence judgments with feedback on calibration is proven effective in improving metacognitive judgments.

3. Given the items selected for inclusion were specifically chosen to ensure that they were ambiguous and performance would be at or close to chance, it seems very unsurprising that confidence ratings were uncalibrated with accuracy? Is this lack of calibration not just a function of the stimuli chosen? To have decent confidence accuracy calibration across the confidence range, shouldn't there be variation not only in the confidence ratings participants can provide but also in their performance on the items? For example, if items are intentionally selected so that there are no items on which participants are 80% accurate, then should never select 80% confidence? Instead, participants should have just selected 50% confidence for all trials because they were performing close to chance? Can the authors please either explain why my concern is wrong (I am open to counterpoints or pushback) or update the manuscript to note this as a limitation and appropriately adjust their conclusions.

Items have been chosen to ensure performance at chance *on average*. However, chance-level performance is not achieved for all stimuli and we have some variance in performance. Indeed, some stimuli were easier to evaluate whereas others were more difficult. The mean success rate, as computed per stimulus, is 51.58% (SD=20.24%). The most difficult stimulus to evaluate has success rate of 6.92% and the stimulus the easier to evaluate has a 93.75% success rate. Thus, we do have a broad range of success rates.

In fact, the success rate at the assessment task has never been a criterion of stimulus selection. We hypothesised that deviations from chance level in evaluating news truthfulness could be explained by the level of ambiguity, rather than by other variables. This was the initial rationale for using stimuli that vary in ambiguity, as we hypothesized that ambiguity is responsible for failures.

It is true that by choosing items that were ambiguous, this led performance to be at or close to chance levels on average. However, it does not necessarily follow that confidence ratings should be uncalibrated with accuracy.

To clarify this point, we have added analyses on the success rate per stimulus in the method section, page 5:

"The resulting dataset had an average success rate, as calculated per stimulus, of 51.58% (SD=20.24%). The most difficult stimulus to evaluate had a 6.92% success rate and the easiest stimulus to evaluate had a 93.75% success rate."

We also return to this point in the discussion section, page 13:

"The news stimuli have been chosen to ensure performance at chance on average, with news that ranged in evaluation difficulty between very easy and very hard."

4. Regarding the confidence elicitation, I am also not entirely sure why the robots explanation and procedure is necessary? Those instructions seem unnecessarily complicated and unclear. Would it not have been sufficient, and easier for participants to understand, to simply tell them that, for each item, an answer with X chance of being correct will be selected. If their confidence is below the threshold, the answer will replace their own, whereas if their confidence is above the probability of the answer being correct, their answer will be kept. Can the authors please clarify why this was necessary and/or note it as a potential limitation.

Our confidence elicitation procedure is based on Karni's (2009) mechanism for eliciting probabilities. This method falls under the category of "probability-matching elicitation procedures," as outlined by

Charness, Gneezy, and Rasochoa (2021). These procedures are valued for their theoretical robustness, particularly in their resistance to deviations from expected payoff maximization and expected utility maximization, that is, they are immune against the different risk attitudes of the participants. Karni (2009) argues that these mechanisms perform better than traditional proper scoring rules, especially when participants must exert effort to provide reliable probability estimates. This elicitation method is incentive-compatible (players maximize their payoff by reporting their true belief) and used as a standard in behavioral economics.

We acknowledge that one potential limitation of such mechanisms is their complexity. If participants do not fully understand the randomization process, their responses may not accurately reflect their true beliefs. To mitigate this, we adopted the approach used by Coffman (2014) and Möbius et al. (2013), which employ a narrative involving robots to provide a more intuitive explanation of the procedure. We closely followed this explanation in our instructions and included a comprehension questionnaire to ensure participants understood that their best interest is to reveal their belief honestly.

Regarding the suggestion to use a simpler mechanism, Karni (2009) provides a compelling argument for the superiority of probability-matching elicitation procedures over proper scoring rules, particularly in contexts requiring significant cognitive effort to generate reliable probability estimates. Schotter and Trevino (2014) also highlight the robustness of Karni's method, noting that truthful reporting is a dominant strategy regardless of risk attitudes, provided that stochastic dominance is respected. Moreover, it is not so clear that a quadratic scoring rule is necessarily easier to understand (Charness, Gneezy & Rasochoa, 2021).

While the instructions may seem complex, our participant pool, sourced from GATE-Lab (Behavioral economics lab at CNRS), is accustomed to tasks involving detailed economic behavior explanations. These participants come from very selective universities and have on average high mathematical and cognitive abilities. This familiarity likely reduces the risk of misunderstanding and enhances the validity of our findings. Moreover, when using such procedures, it is standard to indicate in the instructions that participants do not necessarily need to understand the procedure in detail and that they can simply consider that it is in their best monetary interest to report their true confidence level. They are also invited to ask the experimenters for additional clarification if they wish.

We provide the details of the comprehension test in the supplementary materials (Supplementary III.3).

We have added the following text to address your point on page 5:

“This elicitation method is incentive-compatible, considered relatively easy to understand⁴⁴ and frequently used in economics^{45,46}. With this elicitation method, truthful reporting is the unique dominant strategy. We adopted an approach used in previous experiments^{47,48} which employs a narrative involving robots, to explain participants its process (Supplementary III.2).”

5. Unless I have missed something, I am not entirely sure why participants would ever want to request additional information if they also wanted to maximize the money they received? I can see a reason to request additional information if that information was provided within the experiment and participants were then able to update their veracity judgments and improve their chance of being rewarded. However, receiving the information after the experiment seems to mean that any request for additional information will reduce the total money they receive? Couldn't participants therefore put the minimum willingness to pay and just search themselves about anything they found interesting or wanted to know more about? Similarly, although I understand the desire to balance the cost of receiving vs. not receiving information, it is even less clear to me why participants would ever pay money to not receive information? Surely receiving the information does not incur any cost even if they do not want it because they could simply choose to ignore the email or not read the information? As with the above points, can the authors please either clarify

if I have misunderstood something or am wrong, or note this as a limitation and adjust their conclusions appropriately.

In the task, participants had to pay for the opportunity to decrease, after the task, ambiguity (or, in some cases, to avoid reducing ambiguity) about the news evaluated. Receiving information after the experiment indeed means that any request for additional information may reduce the total money they receive, if the corresponding trial has been randomly drawn for implementation. Thus, if participants were only motivated by their selfish monetary interest, they should never pay for obtaining additional information or for avoiding additional information.

The fact that that most participants *did* pay shows that they attribute an *intrinsic value* to receiving or not receiving the news (the value of knowing more about a fact or avoiding being exposed to the debugging of a news that they want to believe as being true, for example). Introducing a cost certainly affects the demand for information, but it should not affect the fact that such demand is correlated with one's confidence about the truthfulness of the news. Moreover, introducing a monetary cost is realistic in the sense that spending time to verify an information is costly to the individuals in the real-world.

You are right that participants could still look themselves online for additional information that may decrease uncertainty. However, for that, 1) they would have to remember the stimuli of interest among the 48 stimuli they evaluated – all of which could be up to 140 characters (Old-Twitter format); 2) Participants would endure an effort cost to find that information that would be higher than the monetary cost in the experiment, and 3) there is an opportunity cost if they decide not to pay during the task for news about which they would like to have ambiguity reduced, for the reason they would have to remember it.

Regarding your legitimate concern about the willingness to pay *not to receive* additional information, recent articles have shown that individuals value ignorance, including in tasks with non-instrumental information (Charpentier, Bromberg-Martin and Sharot, 2018; Sharot et Sunstein, 2020; Hertwig et Engel, 2016; Persoskie, Ferrer et Klein, 2014; Golman, Hagmann et Loewenstein, 2017; Kobayashi et al., 2019). The literature in behavioral economics has shown that people are even willing to pay for avoiding being exposed to some information. For instance, in Eil and Rao (2011) participants are willing to pay to avoid receiving unfavorable information about their appearance or intelligence; Ganguly and Tasoff (2017) have shown that people are likely to pay to avoid receiving, via testing, information about herpes infection. Whereas information in the latter experiments likely has an ego-relevant dimension and emotional and instrumental utility rather than a cognitive utility, Charpentier et al. (2018) have shown that people are willing to pay for ignoring non-instrumental information, as a function of the expected valence of knowledge.

We have emphasized the motivations for eliciting the Willigness-To-Pay both for receiving and for avoiding to receive additional information in the Materials and Methods, page 6:

“ This choice of design was motivated by the findings in the literature that people may wish not to receive information that decreases the uncertainty they would rather maintain (i.e., valuation of ignorance⁴⁸⁻⁵³). ”

This being said, we acknowledge that to save money, participants could decide to ignore the email or not read the information instead of paying for not receiving it in the first place. Therefore, we certainly minimize the distaste for additional information. This is a potential concern about the design that we could not solve with manipulating information from written news media because to make sure that the received news was read, we would have to condition the reward on answering questions about the news. However, in this case, the utility of information would have been instrumental, rather than cognitive, which would be adversarial to testing our hypotheses. We have emphasized this aspect of the design in the discussion (see p. 14):

“To fully contextualize our findings, it is important to acknowledge a limitation of the design. Participants could choose to ignore the post-task email or not read the additional information it might contain, rather than paying to avoid receiving additional information. This behavior could not be

controlled and stems from the manipulation of written news media. If the reception of information had been endogenous to the task, verifying that it was read would have required conditioning rewards on responses to questions about the news. However, this would have shifted the utility of the information from cognitive to instrumental, which would be adversarial to testing our hypotheses.”

6. I have some concerns about the pretesting that was conducted to select items for inclusion. Specifically, the authors state that 55 independent raters assessed the 210 headlines, but only assessed 42 headlines each. That means that each headline was only rated by 11 raters, which seems to be too few to ensure that you have reliable ratings? The average random raters’ correlation coefficients reported in Supplementary II also seem quite low (0.54 for content imprecision and 0.59 for desirability, consensuality seems decent at 0.81).

Each stimulus was indeed rated by 11 raters. We acknowledge that this number is not as high as we would have liked it to be. This number was imposed by restrictions at the time of the pre-test, during COVID-19, by the low number of French people taking part to online experiments and by the difficulty to recruit new French subjects at the time of the pre-test.

However, recall that the selection of stimuli was based on the news themes categorization by the raters. The ratings of content imprecision, consensuality, and desirability were not used to select the stimuli for the task. From those ratings, we computed scores of Intra-Class Correlation Coefficients.

The literature on the use of ICC, in particular in medicine, casts light on rules of thumb to choose a number of raters and to interpret ICC scores. The minimum number of raters recommended goes from 3, with at least 30 heterogeneous samples (Koo & Li, 2016), up to 20 (Mondal et al., 2024). In our study, we had 11 raters and 42 heterogeneous samples. Under such conditions, “ICC values less than 0.5 are indicative of poor reliability, values between 0.5 and 0.75 indicate moderate reliability, values between 0.75 and 0.9 indicate good reliability, and values greater than 0.90 indicate excellent reliability” (Koo et Li, 2016). We added those numbers in the Materials and Methods, page 5, by giving more context about the ideal number of raters and the interpretability of ICC scores:

“We computed the scores of news content imprecision ($M=5.28$, $SD=1.32$) and propensity to polarize ($M=6.41$, $SD=1.62$) using Intra-Class Correlations, with 11 raters and 42 heterogeneous samples. The recommended number of raters ranges between 3, with at least 30 heterogeneous samples³⁸, and 20³⁹.”

We also reminded the readers to be cautious when interpreting the ICC scores in the Supplementary materials (Supplementary II):

“ICC values below 0.5 usually indicate poor reliability, whereas values between 0.5 and 0.75 indicate moderate reliability, values from 0.75 to 0.9 indicate good reliability, and values above 0.90 indicate excellent reliability.²”

and in the Discussion section, page 13:

“Caution should be exerted when drawing conclusions from the measures of ambiguity. ICC values indicated moderate reliability of raters on the scoring of news content imprecision and good reliability for the propensity to polarize.”

7. Additionally, is noted in Supplementary II that desirability was predictive of requesting more information. However, only imprecision and polarization were considered within the main body of the paper. Can the authors please either also report the impact of desirability and/or explain why this would be inappropriate.

We did not focus on desirability in the paper, as computing the correlation of reception choices with the desirability rating was only intended to be a stimulus quality check.

8. The organization ratings seem to confound familiarity and closeness. It seems perfectly plausible that someone could be very familiar with an organization and strongly oppose them, which may make averaging inappropriate. Can the authors report the reliability for their combined measure and/or better justify combining them.

We computed the adhesion to organizations as a score of social distance/social proximity to organizations by aggregating six responses to the questions on familiarity, closeness of values, and liking.

We decided to compute the adhesion to organizations as a proxy for beliefs by adapting measures of social distance between individuals. In our context, a closer social distance would mean more involvement in the concerns related to the themes. To that purpose, as in Liviatan, Trope and Liberman (2008) we measured closeness and liking. Contrary to Liviatan, Trop and Liberman (2008), as it may be difficult to interpret what it means for a human being to be similar to an organization, we replaced similarity with familiarity with the organization. This was based on the premise that the more individuals would be involved in the cause on a particular theme, the more these individuals would be aware of the different actors operating in that theme.

Consequently, we assumed that beliefs could be approximated by aggregating whether one is familiar with an organization, whether one likes the organization, and whether one feels one's values are close to those of the organization. We also assumed that familiarity, likeness, and closeness of values of one's relatives influence one's beliefs. We computed an aggregated score accordingly, that we called *social distance*. To check those assumptions against the effect of each isolated item, we computed linear models of veracity judgment, success in estimating veracity and reception choices. We modelled each variable with either the main effects of the six responses, or the aggregated response.

We found inconsistent patterns, where no isolated item could consistently predict any variable better than the aggregated response. In the case of veracity judgment, predictors in the linear models were frequently non-significant. In the case of the success in estimating veracity, we found that the aggregated response was the most consistent predictor. In the case of reception choices, we found that the closeness of values between the relatives and the organization, and the relatives' likeness of the organization, were more significant predictors than those same values for the subjects themselves. Overall, we found that the aggregated responses were not outperformed by the single items, and that the aggregated responses failed to significantly predict the variables when the isolated responses failed as well to significantly predict the variables. Those results motivated us to use the aggregated response as the most accurate proxy for beliefs.

We clarified the rationale for employing those measures page 5:

“As a proxy for beliefs, we adapted measures of social distance between individuals to the relationships individuals may maintain with organizations. In our context, a closer social distance would mean more involvement in the concerns related to the themes. To do that, as in ⁴², we measured closeness and liking. Unlike for the latter, as it may be difficult to interpret in the context of a human being's relationship to an organization, we replaced similarity with familiarity with the organization. This approach was based on the premise that the more an individual is involved in the concerns related to a particular theme, the more that individual will be aware of the different actors operating within that theme.”

9. In the discussion (Page 9, paragraphs 2 and 3) it is noted that participants overestimated the amount of true news and several potential explanations are provided. One important potential explanation that should be considered is that the vast majority of information that most people encounter is true (Acerbi et al., 2022; Guess et al., 2019). Therefore, having a baseline assumption that information is true may also be rational based on prior experience. The data collected

regarding the perceived percentage of fake news from the various sources may also provide some insight into this (i.e., if they perceive a small percentage of information coming from those sources to be fake news, and particularly for the overall question about the internet in general, that may help explain why they generally assumed information was true and overestimated).

We thank you for these references that we have added to our discussion along with an expansion of the reasons why subjects may have a bias for declaring news as being true (see page 13):

“It may also be that participants’ held a baseline assumption that information is true, given the prevalence of true information people encounter⁶⁷⁻⁶⁹.”

10. Finally, I request that the authors add a statement to the paper confirming whether, for all experiments, they have reported all measures, conditions, data exclusions, and how they determined their sample sizes. The authors should, of course, add any additional text to ensure the statement is accurate. This is the standard reviewer disclosure request endorsed by the Center for Open Science [see <http://osf.io/hadz3>]. I include it in every review.

We have added this statement in a dedicated section called *Disclosure statement*:

“The authors confirm that all conditions, data collection procedures, data exclusion procedures and sample size determination procedures have been reported. All collected data and questionnaires have been reported either in the manuscript or in the supplementary materials. The instructions to the task and the questionnaires, translated into English, are all available in the supplementary materials. Exposition to information has been collected in the post-task questionnaire but not analysed due to being outside the scope of this manuscript.”

Responses to Reviewer #3

The paper includes a study evaluating participants' perception that a series of ambiguous and apolitical information is true or false and their confidence about it. The paper investigates the link between truth perception, confidence and willingness to learn more about the information.

While the empirical content might seem modest - a single study with a modest sample size, the study is quite ambitious and was developed with great care and attention (e.g., pilot study, pretest, attempts to control for a number of variables). For example, the news used span different domains and the sentiment/valence of the different pieces was assessed, along with people's perception of key aspects of the news (e.g., potential to divide, ambiguity). So, well done for this.

However, because of the complexity of the study, the clarity of some of the points is hard to grasp – including important methodological choices – and their implications. For example, regarding the complex way to implement incentives (described on page 11 and 12) and the way to operationalise information seeking/avoidance. This lack of clarity together with a complex design also means that some of the analyses seem to be tentative and explorative rather than hypotheses driven (e.g., using values from a pre-test to predict participants' behaviour).

Thank you so much for your support and positive comments about our study. We are grateful for your comments and suggestions that have helped us considerably revise our manuscript. We hope we have addressed your points satisfactorily. In our responses below, we use bold font whenever we quote from your letter, and we use standard font for our responses, and italicized fonts for quoted text from the revised manuscript.

Method

1. You report running a sentiment analyses on the materials and report that those were mostly neutral – but a look at the distribution of sentiments show differences in the distribution of negative and positive emotions – the false one had a wider range of negative ratings and the real one had a wider range of positive emotion. Could this maybe be mentioned and addressed in the manuscript?

Thank you for raising this point. We have emphasized in the revised manuscript that the distribution of false information rated as negative by the sentiment analysis is wider than the distribution of true information; whereas we found the opposite for information rated as positive (see page 5):

“The distribution of false information, rated as negative by the sentiment analysis, was broader compared to the distribution of true information. Conversely, we found the opposite trend for information rated as positive.”

We also addressed this point in the discussion section, page 13:

“Although 93.75% of the stimuli were predominantly categorized as neutral by our sentiment analysis, we observed a slight skew in emotional valence. False information exhibited a broader range of negative sentiment ratings, while true information showed a wider range of positive ratings. This subtle difference aligns with literature suggesting that false information often evokes more negative emotions¹. Emotions may promote belief in fake news⁶⁰. Despite this, the overall sentiment distribution remained largely neutral, indicating that the skew did not impact the neutrality of the materials.”

2. Could you explain why using the ratings or organisation as an assessment of World view and not a more classical measure? Also, you mention 12 organisations and 6 judgments – did people judge pairs of organisations? Or did they judge a randomly selected set?

We computed the adhesion to organizations as a score of social distance/social proximity to organizations by aggregating six responses related to familiarity, closeness of values, and liking.

The term *social distance* refers to the concept of Psychological distance, taken from the Construal Level Theory (CLT), as the degree to which events are directly experienced (Lammers, 2012). A greater distance towards an object usually means less involvement with that object (Liberman, Sagristano, & Trope, 2002; Trope & Liberman, 2012). People tend to consider interpersonal relationships as spatial relationships, hence the notion of social distance (Matthews & Matlock, 2011; Yamakawa, Kanai, Matsumura & Naito, 2009).

In the CLT, social distance measures the space between two individuals or two social groups. For example, if people within a group feel close to another group, or feel similar to that other group, or feel like they can relate to that other group, they would be exhibiting a close social distance (Trope & Liberman, 2012; Matthews and Matlock, 2011; Liviatan, Trope, Liberman, 2008). In Liviatan, Trope and Liberman (2008), participants were asked to indicate how similar a target was to themselves and how close they felt to the target. They were also asked to rate their liking the target.

We decided to compute the adhesion to organizations as a proxy for beliefs by adapting measures of social distance between individuals. In our context, a closer social distance means more involvement in the concerns related to the themes. To do that, as in Liviatan, Trope and Liberman (2008) we measured closeness and liking. Contrary to Liviatan, Trope and Liberman (2008), as it may be difficult to interpret what it means for a human being to be similar to an organization, we replaced similarity with familiarity with the organization. This was based on the premise that the more individual would be involved in the concerns related to a particular theme, the more that individual would be aware of the different actors operating in that theme.

Participants evaluated four organizations per theme. These organizations could be intuitively categorized as pairs holding opposing views on a given theme. For example, whereas WWF and Greenpeace can be regarded as organizations that endorse the existence of human-caused climate change, NIPCC and Climato-sceptiques do not endorse the existence of human-caused climate change. The organizations were presented to each participant in a random order.

We clarified the rationale for employing these measures page 5:

“As a proxy for beliefs, we adapted measures of social distance between individuals to the relationships individuals may maintain with organizations. In our context, a closer social distance would mean more involvement in the concerns related to the themes. To do that, as in ⁴², we measured closeness and liking. Unlike for the latter, as it may be difficult to interpret in the context of a human being’s relationship to an organization, we replaced similarity with familiarity with the organization. This approach was based on the premise that the more an individual is involved in the concerns related to a particular theme, the more that individual will be aware of the different actors operating within that theme.”

We clarified the randomized presentation order page 5:

“Organizations were presented to participants in a randomized order.”

3. I might have missed this bit of information, but I was not sure where the data was collected, whether it was done in the lab or from anywhere?

We apologize for the lack of clarify of our design description. The data has been acquired online, via a platform named Testable.org. Participants to the experiment were students from business schools and engineering schools regularly registered in the GATE-Lab pool of experimental subjects, in Lyon, France. This has been highlighted in the revised manuscript on page 4:

“Participants to the experiment were students from business school and engineering schools regularly registered in the GATE-Lab pool of experimental subjects, at the University of Lyon, France.”

4. To measure confidence, why not asking participants about confidence? I can see how a probability question is information (what are the chances that this information is true vs. false) but a score ranging from -100 to 100 might seem a bit counterintuitive, because it is not a linear indicator of confidence (it is a reverse U shape).

The scores ranging from -100 to 100 are indeed meant to capture confidence on a different scale. In our study, each news item is evaluated on a scale of 1 to 100 or -100 to -1, which maintains a linear representation of confidence in each direction. There is no inherent non-linear relationship between the scales used and confidence levels.

5. I did not quite understand the reference to the karni procedure. The paper cited did not include evidence that incentives improved probability evaluation.

The Karni (2009) procedure is widely recognized for its robustness in probability elicitation, particularly due to its theoretical underpinnings that resist deviations from expected payoff and expected utility maximization (that is, it is not biased by the individuals' different risk attitudes). The procedure's effectiveness is discussed in terms of its incentive compatibility and its superior performance compared to traditional proper scoring rules. Although the specific paper may not directly address incentive improvements, Karni's method is supported by a broader literature on probability-matching methods (e.g., Schotter & Trevino, 2014) which underscores its advantages in various contexts.

We emphasize these points pages 5 and 6:

“This elicitation method is incentive-compatible, considered relatively easy to understand⁴⁴ and frequently used in economics^{45,46}. With this elicitation method, truthful reporting is the unique dominant strategy. We adopted an approach used in previous experiments^{47,48} that employs a narrative involving robots to explain the process to participants (Supplementary III.2).”

6. The text page 11 line 439 describes the involvement of robot that does not seem clear. What is the rationale for that? Why not saying something simpler: If your response is correct / better than that of an artificial intelligence?

The use of robots in our instructions is based on the approach by Coffman (2014) and Möbius et al. (2013) that simplifies the understanding of the randomization process. This narrative helps ensure that participants grasp the elicitation mechanism effectively. A simpler description might not convey the complexities involved in probability matching as effectively. Our approach aims to be as transparent as possible while adhering to established methods, which is why we included the detailed instructions supported by a comprehension test (see Supplementary III.3).

We emphasized this aspect in the design page 6, as already cited in response to your previous point:

“This elicitation method is incentive-compatible, considered relatively easy to understand⁴⁴ and frequently used in economics^{45,46}. With this elicitation method, truthful reporting is the unique dominant strategy. We adopted an approach used in previous experiments^{47,48} that employs a narrative involving robots to explain the process to participants (Supplementary III.2).”

7. For example, I was not sure what was meant by “if the randomly drawn robot had an accuracy level higher than their own reported degree of confidence, we would take the robot’s answer into account”. Sorry, this point is important but unclear.

Each robot is associated with a value ranging from 1 to 100. The value corresponds to the robot's level of accuracy, that is, the probability that it provides the correct answer. If the value of the robot is higher than the participant's reported degree of confidence, we would select the response provided by the robot.

This response would have a probability to being correct equivalent to the value it has been assigned. Otherwise, If the value of the robot is lower than the participant's reported degree of confidence, the participant is rewarded only if his or her response was correct. This procedure guarantees that it is always in the participant's best interest to report his or her best and most sincere response.

To quote Karni in his seminal paper (2009): "*The elicitation mechanism selects a random number r from a uniform distribution on $[0, 1]$ and requires the agent to submit a report, $\mu \in [0, 1]$, of his subjective probability assessment of the event E . The mechanism awards the agent the payoff $\beta := x_{Ey}$ if $\mu \geq r$ and the lottery $\ell(r \times y)$ if $\mu < r$.*"

We clarified the mechanism, page 6, by replacing the former explanation of the rewarding mechanism with a new explanation.

Former explanation:

For each selected trial, one robot out of 100 robots was randomly drawn. To each robot was associated an accuracy level between 0 to 100, corresponding to the probability of this robot to provide the correct answer. Participants were aware that if the randomly drawn robot had an accuracy level higher than their own reported degree of confidence, we would take the robot's answer into account; otherwise, we would take the participant's answer into account. Each correct veracity judgment in these eight trials was paid 50 Experimental Currency Units (ECU), with 100 ECU worth \$2.

New explanation:

"To be more specific, for each selected trial, considering the participants have reported their confidence $\mu \in [1, 100]$ regarding their judgment of the news veracity, the elicitation mechanism selected a random number r from a uniform distribution on $[1, 100]$. If $\mu \geq r$, the participants earned a payoff $\beta := 50$ ECU if their veracity judgment was correct and a payoff $\beta := 0$ ECU if their veracity judgment was incorrect. If $\mu < r$, the payoff was determined by a lottery $(r, 50 \text{ ECU}, 0 \text{ ECU})$. Participants understood that if r exceeded their reported confidence μ , the outcome would rely on the lottery; otherwise, their own judgment would be used. Participants were informed that truthful reporting was in their best interest. Each correct veracity judgment in this context earned a reward of 50 ECU, with 100 ECU valued at \$2. In the narrative, r corresponded to the accuracy level of a robot randomly drawn from a pool of 100 robots."

8. Assuming my interpretation of the incentive system above is correct, do you think this set up could have incentivise people to give extreme ratings (meaning close to -100 or close to +100, thereby uncalibrating people's judgment instead of making them more accurate? Why not rewarding based on the closeness to being correct? (e.g., a score of +100 for a real information = score of 50 ECU – score of +50 for a correct news = 25 ECU, with a penalty for scores that are wrong to incentivise against going extreme).

The Karni method has several advantages over other methods. The procedure is incentive-compatible, that is, participants have a financial incentive to reveal their true beliefs (and so, not to choose the extremes unless they truly believe in the extreme value). Participants maximize their expected payoff by being honest about their beliefs. Another advantage is that the method does not require assumptions about the utility function: it is invariant to heterogeneous risk preferences (Charness, Gneezy and Rasocho, 2021). Another advantage of this method is that it is grounded in decision theory with solid theoretical foundations. Some studies have found a central tendency bias when using more complex elicitation procedures (Birnbbaum, 1992, Andersen et al., 2006, Harrison et al., 2007), but this does not apply to Karni (see Gangadharan et al 2024, section 5.3.3). See also Schlag et al. (2024), Coutts (2019) and Charness, Gneezy and Rasocho (2021).

9. Could you explain how veracity was scored? Was it the case that any score between 1 and 100

for a true information was categorised as correct? Could the response scale with both veracity and 0-100 and false on the -100 to 0 side on the same single scale lead people to the perception that they answered about the extent to which the fact was true or false? Which would not quite be a measure of confidence? Could you also clarify how 0 was coded?

From a methodological angle, any score between 1 and 100 for a true information was indeed categorised as correct. The mechanism elicits participants' beliefs about the state (true or false) of the information. This corresponds to the probability, as perceived by the participant, that their response (true or false) is correct. Subjects are therefore asked to give a measure of certainty of their own response, and not a measure of certainty of the category (true/false) of the news.

The mechanism asks participants to choose a direction to declare whether the information is true or false, then employs the slider with more precision to declare a confidence in their own answer. The fact that subjects declare the probability of their choice being correct aligns with the robot having an accuracy degree between 1 and 100, and the mechanism choosing the robot's answer if the robot is more accurate than the subject is certain.

It is possible that the response scale we used may have led people to the perception that they answered about the extent to which the fact was true or false. However, eliciting beliefs about one's own judgment maps to the notion of confidence in one's own judgment. As such, we could always transform the evaluation back into a classic scale: neither true nor false = 50% confidence in judging the news as true, 1% chance that it is true = 51% confidence in judging the news as true, etc. The answer with a score of 1 = 51% confidence in judging the news as true.

The Karni probability elicitation mechanism can elicit beliefs about the veracity of the news, meeting the design requirements. Note that a response with a confidence score of 0% is not permitted during the task.

We emphasized this point page 5:
"Participants could not respond with 0."

10. Could you explain why adding a willingness to pay measure of the information sought (or refused) – and not simply asking people if they wanted to read more about the issue? Did you think that a cost would change their answers?

We believe an incentivization mechanism that penalizes participants' deception about their preferences was necessary to obtain their (true) preferences for information. This is the reason we chose to elicit preferences for information with a standard bidding mechanism such as the Becker–DeGroot–Marschak method. The BDM method is known to elicit the values of participants for products. By basing the method on a non-hypothetical question (in the current study, participants pay with their initial endowment), we elicited the participants' true value they attribute to receiving additional information. Using a bidding mechanism instead of relying on stated preferences, such as ordinal ranking (i.e., simply choosing to receive or not to receive), reduces the likelihood that participants lied in their responses, for self- or social-image concerns, or any reason that is unobservable to the experimenters. Incentivizing decisions is also a requirement in behavioral economics.

We emphasized this point page 7:
"Using a bidding mechanism such as the Willingness-to-Pay instead of relying on stated preferences, such as ordinal ranking (i.e., simply choosing to receive or not to receive), reduces the likelihood that participants did not provide sincere responses."

11. For participants who did not want more information, if they did not “pay” enough for their decision, then, did you send people the information despite their choice? If that would be the case, it seems it would be an ethical issue that should be mentioned and discussed. This seems to be what is implied in the text: “had to report how much they were 449 willing to pay, between 0 and 25

ECU of their 200 ECU initial endowment, to have their decision 450 implemented (i.e., to receive or not receive further information), [...] If the WTP was lower than the price, no deduction was operated and the option the participant did not choose was implemented.”

This is indeed the principle underlying the BDM mechanism: if participants did not pay enough to avoid receiving more information, we would send the information despite their choice. More specifically, participants received more information about the news, contrary to their decision. Participants were perfectly informed about the mechanism which ensures that it is always possible to avoid receiving information for sure by paying the maximum amount.

In addition, because we exposed participants to both true and false news, we were bounded due to ethical reasons to send each participant a debunk of each news they evaluated. Participants received by mail a list of news with a mention “True” or “Fake” next to each of them. In addition, participants received investigation articles based on the implementation mechanism.

This choice of design was motivated by the findings in the literature that people may wish not to receive information that decreases the uncertainty they would rather maintain (i.e., valuation of ignorance, Charpentier et al., 2018). The first paper we know that addressed this behavior is Dana, Weber & Kuang, (2007).

We have clarified these points in the design section, page 6:

“Participants were informed about the mechanism and it was always possible to avoid receiving information for sure by paying the maximum amount. If participants did not pay enough to avoid receiving more information, participants received more information about the news, contrary to their decision.

[...]

This choice of design was motivated by the findings in the literature that people may wish not to receive information that decreases the uncertainty they would rather maintain (i.e., valuation of ignorance⁵⁰⁻⁵⁵).”

12. Also, do you think that people’s greater willingness to pay to get information (vs. not to get it) could be motivated by the fact that they would not have to read it – providing a low willingness to pay avoid the risk of forgoing some of their endowment and they would simply delete the email?

Regarding the willingness-to-pay to receive, it is likely that the willingness-to-pay is lower for not receiving information than for receiving it, as one can simply choose not to read the information. This is acknowledged in the text.

Regarding the willingness-to-pay not to receive, participants could also decide not to pay not to receive the information, ending with the outcome of receiving the information and choosing not to read it, for instance by deleting the email.

In any case, there is a precedent in the literature that people pay either to receive and not to receive information (Charpentier, Bromberg-Martin and Sharot, 2018; Sharot et Sunstein, 2020; Hertwig et Engel, 2016; Persoskie, Ferrer et Klein, 2014; Golman, Hagmann et Loewenstein, 2017; Kobayashi et al., 2019) and we observed that our participants did as well, with a significant difference.

13. The Supplementary materials include a lot of information. It would be useful for interested readers to have an index. I struggled for example to find some of the information referred to in the manuscript in the SM (e.g., I could not find Supplementary VI.2, about the comparison of outcomes across data collection waves. – I think it might actually be 3. In SM)

We have made the requested changes and we apologize for the oversight.

14. Page 2, line 56 – it is not clear what the characteristics that increase ambiguity are.

We apologize for the lack of clarity in our previous version of the manuscript. We changed the text to clarify these characteristics, by specifying, page 2:

“These two characteristics, imprecision and polarization, increase ambiguity^{9,10}, amplifying the difficulty to discern between true and false information.”

15. Could you provide a rationale for using response time? It is not clear what it is supposed to measure – e.g., intuitive processing?

The response time measure is a proxy for cognitive reflection, known for its ability to account for individual differences in information evaluation. This is now specified page 10:

“(...) response times, likely reflecting cognitive reflection”.

Literature review/introduction/Discussion

16. There is quite a bit of research about confidence and belief quality in the literature – but those findings are described as focusing on “perceptual task”. I am not sure what that means. People are quizzed with factual questions to which they provide answers that are either correct or not and then, they are typically asked how many correct answers they provided overall. It is quite similar to what is being done in the article and could be better presented.

We have rephrased our statement as its previous formulation indeed lacked clarity.

The literature described focuses on decision-making tasks where participants form beliefs about the true state of visual stimuli. The present research is a decision-making task where participants form beliefs about the true state of verbal stimuli.

We made the expected clarification on the matter on page 14:

“In tasks with perceptual information, the state of the world is directly accessible and potentially identifiable with enough time to accumulate evidence. In the case of textual information, prior knowledge could theoretically aid in the evaluation of stimuli. However, active engagement and motivation to consider prior knowledge are necessary for effective evaluation⁵. Such processes do not occur routinely during comprehension of textual stimuli. A key factor that may influence whether individuals engage in careful evaluation is their beliefs about their own susceptibility to misinformation, pointing to an additional metacognitive dimension⁷⁷.”

17. It might be helpful to give an example of news in the manuscript – as I think that the fact that the choice of choosing maximally ambiguous non-partisan content (for which you expected chance level veracity judgment) might also explain the lack of link between confidence and accuracy in veracity judgments. This is a potential boundary condition that could be mentioned in the discussion. It does not take away the value of the results – it simply limits the potential situations to which it could apply. The discussion does include a description of the neutrality of the news – but does not include the implication of this for the findings.

We included examples of news items in the manuscript to clarify our approach and the implications for our findings page 3:

“For example, ‘Some well-known athletic committees still deliberately maintain certain sports-related discriminations,’ ‘In 2018, greenhouse gas emissions decreased by 4% in France for the first time’, and ‘The most common voting system in Europe is the “preferential vote,” which involves ranking candidates in order of preference’.”

We understand your concern regarding the potential impact of our choice of news content on the observed lack of correlation between confidence and accuracy. We first want to clarify that our study

utilized a large range of news items with varying degrees of ambiguity, rather than focusing solely on maximally ambiguous contents. As illustrated above, the items were selected to reflect varying levels of content imprecision and polarization, ensuring a realistic and diverse range of news difficulty. This approach was intended to capture a spectrum of evaluation difficulties and to avoid the potential biases associated with extreme cases of ambiguity.

We had not previously included the distribution of average success rates across stimuli in the manuscript, but we agree that this information could be valuable for understanding the confidence-accuracy relationship. The average success rate for evaluating these news items was 51.58% (SD=20.24%), with the most difficult item having a 6.92% success rate and the easiest having a 93.75% success rate. This variability in difficulty is reflective of a broad range of ambiguity, which may help explain why the confidence-accuracy relationship was flat.

As discussed in our manuscript, the observed dissociation between confidence and accuracy is consistent with findings from perceptual decision tasks, where confidence often correlates strongly with accuracy. However, this relationship can weaken with increased task difficulty, as seen in our study. This pattern may be specific to the evaluation of uncertain news, where the lack of clear information complicates the accuracy of judgments and the correlation with confidence.

Whereas it is difficult to make interpretations on the neutrality of our stimuli, as assessed by our exploratory sentiment analysis, we addressed the implication of this finding page 13:

“Although 93.75% of the stimuli were predominantly categorized by our sentiment analysis as neutral, we observed a slight skew in emotional valence. False information had a broader range of negative sentiment ratings, while true information showed a wider range of positive ratings. This subtle difference aligns with the literature suggesting that false information often evokes more negative emotions¹. Emotions may promote belief in fake news⁶⁰. The overall sentiment distribution remained largely neutral, indicating that this skew did not impact the neutrality of the materials.”

18. In the paper, uncertain is sometimes used instead of ambiguous. I am no sure this is the best terminology. For example, the sea level might rise 2 meters – is uncertainty information – but it is not what I would call ambiguous.

We define ambiguity as characterizing situations where the probabilities of different outcomes are unknown. It involves decision-making in the absence of precise information on the likelihood of different events. This is a particular form of uncertainty (the other form of uncertainty being risk which characterizing situations where the probabilities of the different outcomes are known). We have revised the manuscript to make sure that it is consistent with this definition. We also modified a sentence on page 3 as follows:

“Here, we were interested in understanding the cognitive mechanisms and the relationships between individuals’ judgment about the veracity of ambiguous news (news for which the probability of being true is unknown) they are exposed to, the confidence in such judgment, and the willingness to seek additional information to better assess news veracity.”

19. “We investigated the relationship between objective accuracy in judging news veracity and 105 confidence in this judgment, controlling for how news ambiguity influences the judgments of news 106 as true or false”. \ The research goal hints that ambiguity is manipulated and this is a bit misleading because at that stage people do not know whether it was the case, or how, but reading carefully the method would show that it is not manipulated. Mentioning that this was not a manipulation but that ambiguity was subjectively assessed by a separate sample (of 55 participants) would be useful. It would also be helpful to also provide the distribution of all the ratings taken in that pretest (imprecision, desirability [which is a measure of information seeking], consensuality/divisiveness and themes).

We acknowledge that we sentence may mislead the reader in interpreting that ambiguity was manipulated the same way we manipulated the news theme. However, before the pre-testing, we deliberately modified some news items and manufactured others to obtain variations in ambiguity.

We modified the sentence as follows on pages 3-4:

“We investigated the relationship between objective accuracy in judging news veracity and confidence in this judgment, controlling for the role of news ambiguity in the judgments of news as true or false. At the time of designing the task, we standardized the material to obtain scores of ambiguity by introducing variations of ambiguity in stimuli. Ambiguity was subjectively assessed by a separate sample of 55 participants during a pre-testing.”

20. Partly related to this, I think quite a bit of space is devoted to the unpredicted interaction effect between confidence and ambiguity on information seeking. I do not put the results into question and I find the authors’ interpretation fair, but this result is given maybe too much credit and prominence in the abstract and in the discussion (e.g., in the abstract: “the level of confidence, although uncalibrated, was the primary driver of the demand for additional information about the news, with lower confidence driving a greater demand, regardless of its veracity judgment. This demand for disambiguating information, driven by the uncalibrated metacognition, was increasingly ineffective as individuals became more enticed by the ambiguity of the news.”). In the abstract, readers are unaware of the way this was measured and so might place too much weight on the value of that finding.

We have toned down the findings above and precised, in the abstract, that it was obtained via SEM.

21. Why keeping the cost of seeking (or refusing) information the same? Information comes at a cost, at least an opportunity cost (i.e., time spent reading), or possibly money (e.g., buying newspaper) – whereas refusing information usually does not?

In natural settings, avoiding information also entail costs (for example, developing strategies to avoid being confronted with some information or situations). The literature has shown that people value ignorance and that people are willing to pay for avoiding being exposed to information with instrumental, hedonic and/or cognitive utility (Eil and Rao, 2011; Ganguly and Tasoff 2017; Charpentier, Bromberg-Martin and Sharot, 2018; Sharot et Sunstein, 2020; Hertwig et Engel, 2016; Persoskie, Ferrer et Klein, 2014; Golman, Hagmann et Loewenstein, 2017; Kobayashi et al., 2019). To compare choices to receive information with choices not to receive information and to control for any scaling or anchoring effect, we decided to keep the cost of seeking or not seeking the same, with null (0) up to large (25) amounts of currency available for decision. Using various costs would have been arbitrary and made the comparison impossible.

We emphasized this point on page 6:

“We kept the cost constant between reception choices to compare them while controlling for scaling or anchoring effects. This aligns with literature showing that people may value ignorance and are even willing to pay not to receive information.”

22. Could the authors provide a distribution of confidence ratings for true and false statements across different themes? (maybe in SM)? Is it possible that participants chose their responses randomly?

We included a distribution of confidence ratings for true and false statements across different themes in the table S10 (Supplementary I.3), page 15.

The confidence elicitation procedure, as described in Karni (2009), incentivizes accurate probability estimates, which discourages random responses. Additionally, the comprehension test included in our instructions (see Supplementary III.3) was designed to verify that participants understood the rating scales and the task requirements. The responses to the Willingness-To-Pay to implement the reception choices indicate that participants declared their beliefs. Consequently, we believe it is unlikely that participants chose their responses randomly.

23. I do not see the analyses with response time and prior beliefs – the paper includes the following: “Alignment of beliefs with news concerns and socio-demographics showed no significant effects 216 on confidence (Supplementary I.3, Table S11), while effects of response times were highly 217 significant and negative ($p < 0.001$).” but Table S11 does not include response time measure.

We apologize for the missing Table S11. It is now included in the Supplementary Materials on page 16.

24. The paper implies that people should seek more information about topics about which they feel less confident – but people could merely be driven by interest. It is a thought provoking question to consider whether people should spend time reading about topics that were chosen to be immaterial to important decisions.

The literature has indeed shown that many factors drive information-seeking and endogenous interest is not the most motivating one, especially regarding information that does not directly help increasing gains or avoiding losses. The current paper addresses the type of news that is the most debated on the internet and social media, namely true and fake news with cognitive utility. Such information is far from helping people maximizing their utility and is not the most prevalent information that people encounter in their day-to-day-life. To mitigate misinformation is a hot topic with many promising solutions in the lab and one could only hope to see translated to the real world, at school and on the internet. We think that the present research especially offers arguments in favour of helping people calibrating their metacognition.

Minor comments

25. I found the mean +- SD formulation a bit confusing – (as a psychologist) – maybe it is common for other disciplines? I find that it seems to suggest that the mean only vary by the SD value, which is obviously not the case. I find the $M (SD = XX)$ or $(M = XX, SD = XX)$ better options.

Thank you for bringing this concern. We acknowledge that using the +- SD formulation can be confusing and have changed accordingly +- for $M (SD =)$ in the entire manuscript and the Supplementary Materials.

26. Maybe a slight rewrite would help to understand what is being predicted and what is used to predict it? “Modelling the success in estimating veracity confirmed that the veracity judgment was a highly significant explaining variable” or maybe here there is a typo and the statement should be “Modelling the success in estimating veracity confirmed that the veracity of the news was a highly significant explaining variable” – which could be made even clearer and simpler as “Analyses supported that the veracity of the news predicted veracity perception”?

We apologize for the misleading sentence as well as the typo in the Supp. Mat reference. We have clarified this by changing the first sentence with the second sentence on page 9:

Former sentence: “*Modelling the success in estimating veracity confirmed that the veracity judgment was a highly significant explaining variable ($p=0.003$), withstanding the inclusion of control variables (Supplementary I.1, Table S6).* “

Modified sentence: “*Analyses supported that the veracity judgment predicted veracity perception ($p=0.003$), withstanding the inclusion of control variables (Supplementary I.1, Table S4).*”

27. In SM these two statements seem to contradict each other: “Fifty-five independent raters ($F=33, M=22$; mean \pm SD age= 26.2 ± 4.78) received the 210 true and false news. Five groups of 11 French-speaking raters evaluated each a subset of 42 news out of the 210.”

We acknowledge that the two sentences seem to contradict each other. In order to clarify the statement, we now write instead “A total of fifty-five independent raters ($F=33, M=22$; mean \pm SD age= 26.2 ± 4.78) evaluated the news. Five groups of 11 French-speaking raters evaluated each a subset of 42 true and false news out of the 210.”

You will find in the SM, page 22, the new sentence:

“*A total of fifty-five independent raters ($F=33, M=22$; mean age= 26.2 , SD age= 4.78) evaluated the news. Five groups of 11 French-speaking raters evaluated each a subset of 42 true and false news out of the 210.*”

28. The correlation between the desirability of information (how much do you want to learn more) in the pretest and information seeking in the main study is described in the SM as weak but it seems could it is more medium ($R\hat{h}=0.349$).

We used the term “weak” in this context as caution, but we happily made the suggested change.

29. All odd ratios < 0.76 – I think you meant > 0.76

We confirm that all odd ratios corresponding to the analysis with a reported value of < 0.76 are below 0.76. This is apparent in Table S7 (Supplementary I.1), for the model of success, with interactions *VeracityTrue:Imprecision* and *VeracityTrue:Polarization* bearing odd ratios, respectively, of 0.75 and 0.62. We acknowledge that the interaction effects reported as raw values in a table alone may confuse the reader. For instance, with *VeracityTrue:Imprecision*, the term indicates how **Imprecision** influences **Success** when **Veracity** is **True**. The odds ratio (OR = 0.75) suggests that as **Imprecision** increases, the odds of **Success** decrease by 25% when **Veracity** is **True**.

For that same reason, to clarify the results, we reported the estimated marginal means along with the main and interaction effects: “*Specifically, success in judging true news increased when their content imprecision and propensity to polarize were at their minimum (minimum/maximum, imprecision odds-ratio = 1.82; polarization odds-ratio = 2.17) Conversely, for false news, success increased with maximal imprecision (minimum/maximum odds-ratio = 0.53) and maximal propensity to polarize (minimum/maximum odds-ratio = 0.22) (all $p < 0.001$).*”

30. Page 9, line 325 – gullibility seems a bit too pejorative? Maybe trust would be a better terminology?

Thank you for your suggestion. The term gullibility stems from a paper from Tomasello, cited at reference 45 (Tomasello, M. (2020). The ontogenetic foundations of epistemic norms. *Episteme*, 17(3), 301-315.) However, we have replaced “gullibility” with “trust”.

Overview of major changes

Manuscript:

- Modified the title
- Updated the authors information
- Modified the abstract
- Moved the Methods section before the Results
- Updated the mean \pm SD for mean (SD=)
- Detailed the p-values instead of reporting “all p” to be in line with the editorial requests
- Modified the figures that presented barplots without individual data to be in line with editorial requests
- Revised the terminology of uncertainty
- Added a Disclosure statement, page 16
- Added a Data availability section, page 16
- Added a Code availability section, page 16

Abstract:

- Specified the results from the SEM, page 1

Introduction:

- Citation corrected (Lichtenstein & Fischhoff, 1980) along with new citation (Callender, Franco-Watkins & Roberts, 2016), page 1
- Specified the notion of ambiguity in the context of the paper page 2
- Specified that variations in ambiguity were introduced when designing the task, page 3
- Corrected a sentence, page 3

Materials and Methods:

- Re-specified the participants recruited, page 4
- Added the minimum and maximum ages of participants to be in line with editorial requests, page 4
- Specified that the study was not preregistered to be in line with editorial requests, page 4
- Added the mean agreement on themes, page 4
- Added analysis of stimuli, pages 4-5
- Recontextualized the ICC analysis by adding the recommendations from the literature, page 5
- Specified the shape of the distributions for the sentiment analysis, page 5
- Justified further the measures of adhesion to organizations, page 5
- Specified that the organizations were randomly presented to participants, page 5
- Moved a sentence on the comprehension questionnaire from page 5 to page 6
- Corrected a sentence to increase comprehension, page 5
- Specified that participants could not respond with 0 in the evaluation phase, page 5
- Justified the incentive mechanism added page 5
- Specified the properties of the Karni procedure, page 6
- Rewrote the explanation of the Karni procedure to improve comprehension, page 6
- Specified the rationale for keeping constant WTP values between choices to receive and choices not to receive, page 6
- Specified the properties of the WTP, page 6 and 7

- Specified data collection to be in line with editorial requests, page 7
- Specified the versions of softwares used for data analysis to be in line with editorial requests, page 7
- Re-specified the estimation of the sample size, page 7

Results:

- Rewrote a sentence on the effect of veracity judgment on the success in estimating veracity, page 9
- Specified that the response times could be interpreted as cognitive reflection, page 10

Discussion:

- Included 3 news as example of the stimuli used in the task, page 13
- Concerns about the use of headlines of varying ambiguity addressed, page 13
- Sentiment analysis discussed, page 13
- References on the truth bias added, page 14
- Changed ‘gullibility’ for ‘trust’, page 14
- Discussed the truth bias in terms of baseline assumption that information is true, page 15
- Insisted on interpreting ICC with caution, page 14
- Discussed the success rate per stimuli, page 14
- Specified the relationship between difficulty in predicting accuracy and news content, page 14
- Re-specified the distinction made between tasks with perceptual information and tasks with textual information, page 14
- Added limitations of the design, such as allowing participants not to read the news received by mail, page 15
- Discussed the metacognition and the potential lack of awareness of susceptibility to misinformation, page 15
- Specified the results from the SEM, page 15
- Reframed the concluding paragraph to prevent readers from mistaking the take-away with policies or intervention recommendations in line with editorial requests, pages 15 and 16

Supplementary Materials:

- Updated the mean \pm SD for mean (SD=)
- Added an index, page 1
- Added a References section, page 45
- Modified the figures that presented barplots without individual data to be in line with editorial requests
- Rewrote the specification of the raters in section II, page 22
- Framed the interpretation of ICC measures in line with the literature in section II, page 23
- Corrected the term ‘weak correlation’ for ‘medium correlation’ in section II, page 23
- Added the translation of the comprehension test in section III.3, pages 30-37

References

- Andersen, S., Harrison, G. W., Lau, M. I., & Rutström, E. E. (2006). Elicitation using multiple price list formats. *Experimental Economics*, 9, 383-405.
- Bago, B., Rand, D. G., & Pennycook, G. (2020). Fake news, fast and slow: Deliberation reduces belief in false (but not true) news headlines. *Journal of Experimental Psychology: General*, 149(8), 1608–1613.
- Birnbaum, M. H. (1992). Violations of monotonicity and contextual effects in choice-based certainty equivalents. *Psychological Science*, 3(5), 310-315.
- Callender, Franco-Watkins & Roberts, 2016, Improving metacognition in the classroom through instruction, training, and feedback
- Charness, G., Gneezy, U., & Rasochoa, V. (2021). Experimental methods: Eliciting beliefs. *Journal of Economic Behavior & Organization*, 189, 234-256.
- Charpentier, C. J., Bromberg-Martin, E. S., & Sharot, T. (2018). Valuation of knowledge and ignorance in mesolimbic reward circuitry. *Proceedings of the National Academy of Sciences*, 115(31), E7255-E7264.
- Coffman, K. B. (2014). Evidence on self-stereotyping and the contribution of ideas. *The Quarterly Journal of Economics*, 129(4), 1625-1660.
- Coutts, A. (2019). Testing models of belief bias: An experiment. *Games and Economic Behavior*, 113, 549-565.
- Dana, J., Weber, R. A., & Kuang, J. X. (2007). Exploiting moral wiggle room: experiments demonstrating an illusory preference for fairness. *Economic Theory*, 33, 67-80
- Eil, D., & Rao, J. M. (2011). The good news-bad news effect: asymmetric processing of objective information about yourself. *American Economic Journal: Microeconomics*, 3(2), 114-138.
- Gangadharan, L., Grossman, P. J., & Xue, N. (2024). Belief elicitation under competing motivations: Does it matter how you ask?. *European Economic Review*, 104830.
- Ganguly, A., & Tasoff, J. (2017). Fantasy and dread: The demand for information and the consumption utility of the future. *Management Science*, 63(12), 4037-4060.
- Golman, R., Hagmann, D., & Loewenstein, G. (2017). Information avoidance. *Journal of economic literature*, 55(1), 96-135.
- Harrison, G. W. (2007). Making choice studies incentive compatible. *Valuing environmental amenities using stated choice studies*, 67-110.
- Hertwig, R., & Engel, C. (2016). Homo ignorans: Deliberately choosing not to know. *Perspectives on Psychological Science*, 11(3), 359-372.
- Karni, E. (2009). A mechanism for eliciting probabilities. *Econometrica*, 77(2), 603-606.
- Kobayashi, K., & Hsu, M. (2019). Common neural code for reward and information value. *Proceedings of the National Academy of Sciences*, 116(26), 13061-13066.
- Koo, T. K., & Li, M. Y. (2016). A guideline of selecting and reporting intraclass correlation coefficients for reliability research. *Journal of chiropractic medicine*, 15(2), 155-163.
- Lammers, J. (2012). Abstraction increases hypocrisy. *Journal of Experimental Social Psychology*, 48(2), 475-480.

- Liberman, N., Sagristano, M. D., & Trope, Y. (2002). The effect of temporal distance on level of mental construal. *Journal of experimental social psychology*, 38(6), 523-534.
- Liviatan, I., Trope, Y., & Liberman, N. (2008). Interpersonal similarity as a social distance dimension: Implications for perception of others' actions. *Journal of experimental social psychology*, 44(5), 1256-1269.
- Matthews, J. L., & Matlock, T. (2011). Understanding the link between spatial distance and social distance. *Social Psychology*.
- Möbius, M. M., Niederle, M., Niehaus, P., & Rosenblat, T. S. (2022). Managing self-confidence: Theory and experimental evidence. *Management Science*, 68(11), 7793-7817.
- Mondal, D., Vanbelle, S., Cassese, A., & Candel, M. J. (2024). Review of sample size determination methods for the intraclass correlation coefficient in the one-way analysis of variance model. *Statistical Methods in Medical Research*, 33(3), 532-553.
- Pennycook, G., & Rand, D. G. (2019). Lazy, not biased: Susceptibility to partisan fake news is better explained by lack of reasoning than by motivated reasoning. *Cognition*, 188(September 2017), 39–50.
- Pennycook, G., Epstein, Z., Mosleh, M., Arechar, A. A., Eckles, D., & Rand, D. G. (2021). Shifting attention to accuracy can reduce misinformation online. *Nature*, 592(7855), 590–595.
- Persoskie, A., Ferrer, R. A., & Klein, W. M. (2014). Association of cancer worry and perceived risk with doctor avoidance: an analysis of information avoidance in a nationally representative US sample. *Journal of behavioral medicine*, 37, 977-987.
- Sharot, T., & Sunstein, C. R. (2020). How people decide what they want to know. *Nature Human Behaviour*, 4(1), 14-19.
- Schlag, K. H., Tremewan, J., & Van der Weele, J. J. (2015). A penny for your thoughts: A survey of methods for eliciting beliefs. *Experimental Economics*, 18, 457-490.
- Schotter, A., & Trevino, I. (2014). Belief elicitation in the laboratory. *Annu. Rev. Econ.*, 6(1), 103-128.
- Trope, Y., & Liberman, N. (2012). Construal level theory. *Handbook of theories of social psychology*, 1, 118-134.
- Tomasello, M. (2020). The ontogenetic foundations of epistemic norms. *Episteme*, 17(3), 301-315
- Yamakawa, Y., Kanai, R., Matsumura, M., & Naito, E. (2009). Social distance evaluation in human parietal cortex. *PloS one*, 4(2), e4360.